## Registered report

psychology

implicit bias, social status, gender, income, education, individual differences

**Author for correspondence:**
Bradley D. Mattan
e-mail: brad.mattan@gmail.com

# A registered report on how implicit pro-rich bias is shaped by the perceiver's gender and socioeconomic status

Bradley D. Mattan[1] and Jasmin Cloutier[2]

[1]Annenberg School for Communication, University of Pennsylvania, 3620 Walnut Street, Philadelphia, PA 19104, USA
[2]Department of Psychological and Brain Sciences, University of Delaware, Newark, DE, USA

 BDM, 0000-0001-5004-7542

Although high status is often considered a desirable quality, this may not always be the case. Different factors may moderate the value of high status along a dimension such as wealth (e.g. gender, perceiver income/education). For example, studies suggest men may value wealth and control over resources more than women. This may be especially true for high-income men who already have control over substantial resources. Other work suggests that low-income men and women may have different experiences in educational contexts compared to their richer peers who dominate norms at higher levels of education. These experiences may potentially lead to different attitudes about the wealthy among low-income men and women. In this registered report, we proposed two key predictions based on our review of the literature and analyses of pilot data from the Attitudes, Identities and Individual Differences (AIID) study (*n* = 767): (H1) increasing income will be associated with increased pro-wealthy bias for men more than for women and (H2) income will also moderate the effect of education on implicit pro-wealthy bias, depending on gender. Overall, men showed greater implicit pro-wealthy bias than did women. However, neither of our hypotheses that income would moderate the effects of gender on implicit pro-wealthy bias were supported. These findings suggest implicit pro-wealthy bias among men and are discussed in the context of exploratory analyses of gender differences in self-reported beliefs and attitudes about the rich and the poor.

## 1. Background

Sometimes subtle [1–3] signs of social status are readily observed in others, ultimately influencing how we evaluate people [4]. With some exceptions [5–9], we generally perceive high socioeconomic

status (SES) in others as a positive characteristic [10–13]. Even at the level of implicit associations, evidence suggests that higher SES (among other status dimensions: see [14]) is associated with increasing positivity [15]. Moreover, greater wealth—a component of SES—frequently elicits positive implicit evaluations [9,16,17]. These implicit status-based biases can have a real-world impact, shaping the evaluations and judgements we make in circumstances characterized by ambiguity and/or time pressure [9,18]. Critically, social status cues are seldom perceived in isolation from other visually salient attributes such as gender. Numerous studies have demonstrated that assumptions about status are ingrained into gendered roles [19–23] and interactions [24–27], such that women are assumed to be subordinate to men. Dovetailing with this work on perceived gender roles, the literature on masculinity suggests that the control over social and financial capital that high status entails is important to men's gender identity [28–30]. Weaving together these diverse strands of research, this registered report introduces and tests the hypothesis that men show greater implicit pro-wealthy bias than do women in a large online sample. However, we also predict that gender-based positive associations with wealth will further depend on the perceiver's own position in the social hierarchy [11]. Specifically, we will examine how this masculine preference for the wealthy is associated with the perceiver's own SES (e.g. income, education), predicting that both income and education will shape men's (versus women's) preferences for the wealthy.

## 1.1. Gender roles and status incongruity

The literature on gender roles provides a helpful starting point for the present investigation of how one's gender and SES shape implicit preferences for the wealthy. The gender roles account starts from the premise that social expectations about what is appropriate for each sex predicts gender differences in behaviour [21]. Specifically, women are expected to be communal and focused on supporting the family, whereas men are expected to be agentic and focused on the public sphere ([31]; but see [32]). Due to their stereotypic agency and greater historical participation in the paid workforce, men are often presumed to occupy superior positions relative to women [20,33–36]. This gender differential in the perceived status is reinforced by a collection of ambivalent attitudes and beliefs about women [37,38]. On the one hand, hostile sexism reflects negative attitudes and stereotypes about women with potential negative consequences irrespective of the context [39–41]. On the other hand, benevolent sexism reflects relatively warmer attitudes towards women based in beliefs that women are less capable than and require the protection of men [37,38]. Despite its association with greater warmth towards women [37–39], benevolent sexism can also have negative consequences for the advancement of women due to implied stereotypes (e.g. low competence/agency) that give these warm attitudes a decidedly paternalistic flavour [22,42]. Indeed, women who violate gender roles by pursuing high-status positions frequently face backlash in the form of negative social evaluations [43–47]. Like women, men have also been shown to face backlash for violating gender roles (viz., by being modest: [44,48]). Taken together, these findings provide support for the status incongruity hypothesis, which proposes that violating status-based gender stereotypes has negative consequences for perceived employability and agreeableness [43,47,48].

Backlash resulting from gender–status incongruity may also have consequences for how men and women of varying socioeconomic positions value cues related to SES, such as wealth. Existing evidence suggests that people generally show implicit bias favouring high-status targets; however, this bias is most pronounced in those belonging to a privileged group [15,17]. Given that gender roles can confer on men a privileged status over women, it follows that implicit pro-rich bias may be greater for men than it is for women. However, it remains an open question whether wealthy or highly educated women may also show an implicit preference for status that is similar to that of their wealthy/ educated male counterparts. In one experiment examining a similar question in the domain of race [15], we found an interaction between perceiver race and SES in predicting positive implicit associations with high-SES people such that status-congruent individuals (low-SES Black and high-SES White participants) showed reliable pro-high-SES bias. However, individuals who were status-incongruent vis-à-vis their race (high-SES Black and low-SES White participants) showed relatively unreliable implicit bias favouring high-SES targets. This may be explained by status-incongruent individuals having ambivalent feelings about status. On the one hand, achieving higher rank is generally desirable (but see [49]). On the other hand, status-incongruent Black and White participants may also experience status-based threats (e.g. denigration, exclusion) from high-status White people who do not see them as equals [50,51], making the experience of being high in rank seem less desirable. Previous work suggests that competing (i.e. ambivalent) associations can result in

attenuated implicit bias [52]. Accordingly, one possibility is that status-incongruent women and men may show ambivalent attitudes about the rich and attenuated pro-rich implicit bias relative to wealthy or highly educated men and poor or high-school-educated women.

## 1.2. Social hierarchy and masculine identity

Complementing the gender roles literature, which focuses on how socially prescribed roles reward the behaviours of men and women, the psychology of gender identity (e.g. masculinity, femininity) focuses more closely on identification with one's chosen and/or ascribed gender. Of considerable relevance to the present project, the theory of precarious manhood suggests that masculinity is distinct from femininity in its intimate relationship with status-related concerns [53]. Unlike femininity, which is generally considered stable and resulting from physical maturity, masculinity is seen as a form of status awarded through achievement of cultural standards of manhood [54,55]. Not only are these standards generally demanding, the status of men is precarious in the sense that it can be lost when one fails to measure up to these standards [48,55]. Restoring one's public image as a man when this image is threatened (i.e. gender threat: [55,56]) is thought to require public action to restore perceptions of control or competence [57,58], among other hallmarks of hegemonic masculinity (see [28–30,59]).

Consistent with the picture that men's status is fragile, it stands to reason that men may be more vigilant towards and desirous of opportunities to advance their own status in comparison to women. Indeed, evidence suggests that men compared to women more readily display and pursue higher status when status is construed in terms of social or economic influence [31,60–63]. Even among early adolescents, boys' preference for popularity has a greater impact on their affiliations with peers than it has for girls [64]. In adulthood, men also tend to report valuing wealth [65–69] and power (i.e. control over resources: [70,71]) more than women.

In sum, findings from the literature on masculinity suggest that one reason why men may evaluate high SES more positively is because SES is communicated easily and publicly through various cues (e.g. posture, voice, clothing: [3,72–74]) that convey control over social and financial capital. Publicly conveying such an image would be important for claiming, restoring and/or maintaining one's identity as a man [53]. Because one's status as a man may be easily challenged, public reminders of high SES (e.g. wealth, professional degrees) could, therefore, be of greater value to men (versus women) occupying any position on the social hierarchy. However, this is perhaps especially the case for men whose only claim to status is their income, which is more fluid than educational attainment.

## 1.3. Gendered preference for the rich based on hierarchical position

To more clearly identify how gender shapes implicit pro-rich bias, this study examined the moderation of this bias by the perceiver's own income and education. Examining the contribution of individual differences like income and education is important because our relative position along these distinct dimensions of status may differentially impact how we evaluate wealth in ourselves and others [5,6,11,15,17,75,76].

### 1.3.1. Income

Despite some evidence that men may desire and value wealth more with increasing income levels [65] and that wealth is stereotypically tied to masculinity [59], few studies have examined how positive evaluations of the rich are moderated by the perceiver's own gender and income. One possibility is that status-congruent individuals (viz., high-income men, low-income women) will show stronger implicit pro-rich bias than status-incongruent individuals (viz., high-income women, low-income men). This would result in a Gender × Income interaction, with increasing pro-rich bias in men and decreasing pro-rich bias in women as a function of increasing income. Greater pro-rich bias among rich men would be consistent with previous work showing greater preferences for high-status groups (e.g. the rich) among individuals who consider themselves members of those groups [17,77]. The attenuated pro-rich bias among status-incongruent individuals may be due to ambivalent feelings about the rich (see [52])—a hypothesis we intend to explore. Based in part upon this literature and pilot data reported below, our first prediction (H1) is that men will show greater pro-wealthy bias than do women, particularly as a function of their own increasing income (i.e. Gender × Income interaction).

### 1.3.2. Education

In contrast with income, it is perhaps less clear how education levels may affect pro-wealthy bias, implicit or otherwise. Although education is associated with income, it can also be considered a reliable index of SES on its own [78] that is readily ascertained in social interactions (e.g. through speech: [72]). Education affords greater occupational opportunities [79–81] and upward economic mobility [82,83]. This is especially the case for affiliates of elite academic institutions [84], which further enhance one's prestige [85,86]. Conceptualized in this way as a relatively public and direct representation of one's socioeconomic rank, one might expect education to function similarly to income, increasing pro-rich bias among those who are most privileged (viz., the highly educated). Beyond the direct status-conveying aspects of education (e.g. [72]), advanced education as a component of SES also has a cultural component [87,88], which may further enhance positivity towards the rich among the well educated. For example, students who pursue undergraduate or graduate degrees tend to experience greater exposure to a high-SES cultural environments (viz., academia: [89,90]), potentially developing more positive attitudes towards wealth and the wealthy over time as a result of their contact with a generally high-SES population (cf. [91,92]).

### 1.3.3. Income and education

We have so far considered the contributions of perceiver income and education to implicit wealth-based bias separately. However, one of the objectives of the present study is to determine whether these two factors may differentially impact the way women and men evaluate the rich and poor. Based on an integration of the literature reviewed above, we offer two accounts of how income, education and gender may together shape implicit associations about wealth. These accounts focus separately on women and men and are, therefore, not mutually exclusive.

#### 1.3.3.1. Cultural fit account

The first account is motivated by the cultural component of education. As mentioned above, pursuing higher education involves immersion in a typically middle- to upper-class environment, which may lead to greater affinity towards the rich by proxy. However, the implications of contact with the rich probably differ depending on the income of the student or the student's family. In low-income but highly educated individuals, pro-wealthy associations brought about by familiarity with upper-class peers may be offset to some degree by difficulties in navigating cultural differences across class lines [89,93–95]. Indeed, research has identified the academic context as a potential threat to the class identity of middle-class or first-generation students at elite universities, resulting in greater stress [96] and need for self-regulation [97]. In sum, the cultural component of education's contribution to status suggests that lower-income individuals may experience a combination of increased positive contact with the rich (e.g. as friends and colleagues) and negative challenges arising from differences in cultural fit within academe from which their richer peers are sheltered. Such negative experiences may serve to attenuate implicit pro-wealthy bias in low-income but highly educated individuals [52]. Notably, this prediction runs counter to the prediction based on the prestige component of education as a direct representation of social rank, at least for low-income individuals. The prestige-based component would predict generally enhanced rather than attenuated pro-rich bias as a function of increasing education.

Extending the cultural fit account into the domain of gender, it seems likely that low-income women may have ambivalent associations with academia not just because of their income but also because of their gender. Although women receive undergraduate and postgraduate degrees in similar if not greater numbers than men [80,98], gender disparities persist in educational environments. For example, in high-status disciplines such as science, technology, engineering and mathematics (STEM), women are marginalized professionally [99–103]. Even outside of STEM disciplines, women may find themselves underrepresented at the most prestigious academic institutions [98] and face additional hurdles in the pursuit of advanced degrees [80]. In the light of this evidence, one possibility is that having both low income and higher education may lead to ambivalence about status and, therefore, greater attenuation of pro-rich bias in women compared to men.

#### 1.3.3.2. Precarious manhood account

The literature on precarious manhood suggests a complementary effect of education for low-income men. Having low income would constitute a potential threat to the masculine image as capable and self-

sufficient [104,105]. For low-income men, education may provide one means of safeguarding against such threats to masculinity in that it can be conveyed publicly (e.g. through speech: [72]). Moreover, unlike other sources of status (e.g. income), education is more stable. Accordingly, having an advanced degree may make one's income less important when it comes to conferring status. For example, if men are low on income, they may nonetheless perceive and present themselves as high in status by downplaying their wealth in favour of education (see [106]). Indeed, this strategy is readily endorsed among some second-generation immigrant groups [107].

If ranking highly on any dimension of SES is sufficient evidence of masculinity, then wealthy and/or highly educated men should consider themselves as part of a common high-status ingroup—men, and may ultimately respond positively towards all other high-status individuals (e.g. the rich). In such a scenario, it is unclear whether having high levels of both income and education might further boost implicit preferences for the wealthy in men relative to women. In any case, the two accounts presented here propose opposing but not mutually exclusive effects of education for low-income men and women. Whereas the cultural account suggests that low income and increasing levels of education may attenuate women's implicit positive associations with the rich, the precarious manhood account suggests that this same combination could enhance men's implicit positive associations with the rich.

## 1.4. Predictions

Based on the literature reviewed above and findings from our pilot data, we offered the following hypotheses. (For a time-stamped record of our stage 1 pre-registered predictions, see https://osf.io/d5s23/.) First, we predicted that income would increase implicit pro-wealthy bias more for men (H1A) than women (H1B). Our second key prediction (H2) was that both income and education would interact in shaping men's (versus women's) implicit pro-wealthy bias (Gender × Income × Education interaction). At low-income levels, men relative to women would show greater pro-wealthy bias as a function of increasing education (Gender × Education interaction). The cultural fit account specifically predicted that greater education would be associated with an attenuation of pro-wealthy bias among low-income women (H2A). Not exclusive of the cultural fit account, the precarious manhood account predicted that increasing education would have the opposite effect among low-income men (H2B). At high-income levels, increasing education may further enhance pro-wealthy bias in both women (H2C) and men (H2D), but it was not clear whether this increase would also depend on gender.

# 2. Method

## 2.1. Secondary data source

All data for the pilot and confirmatory analyses for this report come from the Attitudes, Identities and Individual Differences (AIID) study [108]. Collected entirely through Project Implicit (https://implicit.harvard.edu/implicit/) between 2004 and 2007, AIID consists of a large ($N \simeq 200\,000$ samples with complete data) and diverse sample of participants who were randomly assigned to complete implicit (Implicit Association Test—IAT: [109]) and self-report measures of attitudes, identity, motivations and cultural perceptions. Our analyses focus on the rich–poor evaluative IAT. The rich–poor dimension was one of 95 different IAT dimensions used in the AIID study. Because each participant completed measures relevant to a single IAT dimension, the anticipated final sample for confirmatory analysis was substantially smaller than the total AIID dataset ($1100 < n < 1200$). In addition to the IAT, participants also completed a subset of individual difference measures and a standard set of demographics items, including gender, income level and education level. In this registered report, we present all measures relevant to our confirmatory and exploratory analyses (see §2.5 Measures). However, a full overview of the AIID study and its measures are available at https://osf.io/atymr/. The present analyses are in response to a recent call from the AIID team intended to promote the use of registered reports using data from the AIID study. To preserve experimenter blinding and promote the use of registered reports [110], Hussey *et al.* [108] split the AIID dataset into stratified subsamples, with approximately 15% of the data available for pilot analyses and approximately 85% of the data withheld for confirmatory analyses. At stage 1 of this registered report (see https://osf.io/d5s23/), we only had access to the smaller sample intended for pilot analyses (see https://osf.io/entbj/ for confirmation letter from C.R. Ebersole). We provide details on the sample used for pilot analyses, our estimate of the sample size to be used for confirmatory analyses, and the final sample size for

confirmatory analyses (data were released upon provisional stage 1 acceptance: see https://osf.io/65stq/ for data release agreement). Data from the pilot analyses were not included in any confirmatory analysis.

## 2.2. Research ethics

Participants voluntarily completed the AIID study, and no inducements or incentives from the AIID research team were used. At the start of the study, participants viewed an introduction screen that included an informed consent agreement stating that participation was voluntary, non-compensated and with minimal risk. The informed consent agreement also stated that responses were confidential and anonymous, protected and analysed in the aggregate, and with information of who to contact in the event of any queries. Also included was the Project Implicit privacy policy. Consent was presumed through participation in the study. Data were collected in accordance with University of Virginia IRB protocol no. 2003017300.

## 2.3. Inclusion and exclusion criteria

Separately for the confirmatory and pilot data, we initially included all participants who completed the rich–poor evaluative IAT ($n_{pilot} = 274$; $n_{confirmatory} = 1264$). Similarly, all inclusion and exclusion criteria reported below apply to both our pilot data analyses and our final confirmatory analyses.

### 2.3.1. Trial-level exclusions in scoring of the Implicit Association Test

$D$ scores for the IAT were computed in line with existing recommendations, which require some trial-level exclusions (see table 4 of [111]). Namely, trials exceeding 10 000 ms were eliminated prior to analysis. Additionally, latencies for incorrect responses were replaced with the block mean RT plus 600 ms. These exclusions were incorporated into the $D$ scores already provided in the AIID dataset. Relative to the original IAT scoring algorithm [109], the updated scoring algorithm (even with the above exclusions) includes more trials by including practice block data and by transforming rather than excluding error latencies. This has been shown to reduce noise from excessively slow responses and from previous experience with the IAT, ultimately increasing the power to detect relationships between the IAT and individual difference measures such as those used in the present study [111]. Thorough details on the IAT scoring algorithm and relevant trial-level exclusions are provided in the section below that describes the IAT measure and its scoring.

### 2.3.2. Participant-level exclusions

Individuals were required to make site-wide user IDs and passwords prior to participating in any study on Project Implicit, with the result that users could be followed across studies, including repeat participations in this study. Although no repeated IDs were uncovered in our pilot or confirmatory datasets, our pre-registered plan for dealing with repeated IDs was to keep only the ID associated with the earliest time and date. Participant-level exclusions were the same for both the pilot and confirmatory datasets.

After IAT $D$ scores were computed (see section below describing IAT scoring), we implemented participant-level exclusions, following the guidelines offered by Project Implicit (see footnote 4 of [112]). The implementation of the criteria listed below aims to reduce the incidence of careless responding in the final dataset. Specifically, we excluded any participants who met any of the following criteria ($n_{pilot} = 46$; $n_{confirmatory} = 195$):

(1) Greater than or equal to 35% of responses under 300 ms in any one practice block.
(2) Greater than or equal to 25% of responses under 300 ms in any one critical block.
(3) Greater than or equal to 10% of responses under 300 ms in critical blocks.
(4) Greater than or equal to a 50% error rate in any one practice block.
(5) Greater than or equal to a 40% error rate in practice blocks.
(6) Greater than or equal to a 40% error rate in any one critical block.
(7) Greater than or equal to a 30% error rate in critical blocks.
(8) In addition to implementing criteria 1–7 that were used by Nosek *et al.* [112], we also adopted a stricter exclusion criterion, removing any participant with greater than or equal to 10% responses

**Table 1.** Distribution of participants from the pilot dataset ($n = 175$) by gender, income level and education level. Numbers within each cell indicate the sum total of pilot participants in that condition.

| gender | income level (USD) | no high school diploma | high school graduate | associate's degree or some college | bachelor's degree | graduate degree or education |
|---|---|---|---|---|---|---|
| women | <$25 000 | 0 | 0 | 12 | 6 | 1 |
| | $25 000–$49 999 | 0 | 5 | 8 | 17 | 3 |
| | $50 000–$74 999 | 0 | 1 | 7 | 10 | 8 |
| | $75 000–$149 999 | 0 | 2 | 7 | 8 | 7 |
| | >$150 000 | 1 | 1 | 2 | 4 | 5 |
| men | <$25 000 | 0 | 0 | 8 | 7 | 1 |
| | $25 000–$49 999 | 1 | 0 | 2 | 6 | 0 |
| | $50 000–$74 999 | 0 | 2 | 3 | 4 | 4 |
| | $75 000–$149 999 | 0 | 0 | 4 | 9 | 5 |
| | >$150 000 | 0 | 0 | 1 | 0 | 3 |

over 10 000 ms in IAT critical blocks. This additional criterion was implemented to exclude participants who were potentially insufficiently attentive during, or confused by, the IAT.

Finally, we excluded all rich–poor evaluative IAT participants who failed to complete all demographic items in our reported analyses in the following order: income level ($n_{pilot} = 49$; $n_{confirmatory} = 296$), education level ($n_{pilot} = 4$; $n_{confirmatory} = 6$) and gender ($n_{pilot} = 0$; $n_{confirmatory} = 0$).

## 2.4. Participants

### 2.4.1. Pilot dataset

After all exclusions (see above), our final sample consisted of 175 participants ($n_{women} = 115$, $M_{age} = 32.4$ years, $Range_{age} = 14$–66 years). The sample was predominantly White (71.4%), but more than one-quarter of participants were minorities: Mixed Race or Other/Unknown (9.7%), Hispanic (8.0%), Asian/Pacific Islander (6.3%) and Black (2.3%). Four participants (2.3%) did not provide their racial/ethnic demographic information. For a breakdown of our sample by gender, income level and education level, see table 1.

### 2.4.2. Confirmatory dataset

Although it was impossible to know in advance the exact number of participants that would meet our exclusion criteria, we estimated based on the pilot dataset that our final sample prior to exclusions would be approximately 1827 participants.[1] This estimate assumed that our pilot dataset prior to exclusions consisted of a stratified sample of 15% the AIID dataset.[2] Because these samples were stratified by Hussey *et al.*, we anticipated a similar demographic make-up in our confirmatory dataset. Based on the 36% exclusion rate in our pilot sample (see exclusions above), we anticipated a final sample of approximately 1167 participants.

After all exclusions (see above), our final confirmatory sample consisted of 767 participants ($n_{women} = 492$, $M_{age} = 32.5$ years, $Range_{age} = 12$–81 years[3]). The sample was predominantly White (76.3%), but

[1]After stage 1 acceptance, it was discovered that this estimate of participants prior to exclusions erroneously included the pilot sample in the estimate total. The correct estimate should have been 1553 participants prior to exclusions and 994 participants after exclusions. Fortunately, the correct estimate for the projected sample size after exclusions was close to the intentionally conservative sample size of 1000 participants that we used for our stage 1 sensitivity analyses.

[2]Because the pilot data were 17.8% of the total AIID data for the evaluative rich–poor IAT, the number of participants prior to exclusions was lower than the corrected projected estimate of 1553 participants prior to exclusions. For this reason, our final sample size fell below our anticipated final sample size for confirmatory analysis.

[3]Exploratory analyses were conducted by excluding 14 participants between the ages of 12 and 18. The rationale for this exclusion was that income for adolescents may not accurately reflect their actual material resources due to dependence on parents or guardians.

**Table 2.** Distribution of participants from the confirmatory dataset ($n = 767$) by gender, income level and education level. Numbers within each cell indicate the sum total of participants in that condition.

| gender | income level (USD) | no high school diploma | high school graduate | associate's degree or some college | bachelor's degree | graduate degree or education |
|---|---|---|---|---|---|---|
| women | <$25 000 | 2 | 6 | 41 | 36 | 14 |
|  | $25 000–$49 999 | 2 | 3 | 43 | 51 | 39 |
|  | $50 000–$74 999 | 3 | 3 | 30 | 37 | 14 |
|  | $75 000–$149 999 | 2 | 1 | 30 | 46 | 38 |
|  | >$150 000 | 2 | 4 | 16 | 14 | 15 |
| men | <$25 000 | 3 | 4 | 21 | 15 | 14 |
|  | $25 000–$49 999 | 1 | 4 | 21 | 25 | 10 |
|  | $50 000–$74 999 | 1 | 3 | 17 | 19 | 18 |
|  | $75 000–$149 999 | 0 | 1 | 15 | 24 | 33 |
|  | >$150 000 | 1 | 2 | 3 | 11 | 9 |

nearly one-quarter of participants were minorities: Mixed Race or Other/Unknown (7.4%), Asian/Pacific Islander (5.7%), Hispanic (5.5%) and Black (3.5%). Ten participants (1.3%) did not provide their racial/ethnic demographic information. For a breakdown of our sample by gender, income level and education level, see table 2.

## 2.5. Measures

Participants in the AIID study completed a number of measures in addition to the rich–poor IAT. We present first all measures involved in confirmatory analyses, followed by additional measures that may be used for supplemental exploratory analyses.

### 2.5.1. Measures for confirmatory analysis

Key independent variables used for confirmatory analyses include participant gender, income level and education level. The IAT $D$ score served as our key dependent measure of implicit pro-rich bias. These measures are described in greater detail below.

#### 2.5.1.1. Gender
Participants were asked to report their sex. Their only response options were male or female. Any participant failing to respond to this item was excluded from all analyses.

#### 2.5.1.2. Income level
The annual income level was assessed on a five-point scale: (1) < $25 000, (2) $25 000–$49 000, (3) $50 000–$74 999, (4) $75 000–$149 999, and (5) > $150 000. All dollar amounts were in US dollars. Participants were also allowed to select 'I don't know' for this item. Participants choosing this option or failing to provide a response on this measure were excluded from all analyses. This variable was $z$-transformed prior to all analyses.

#### 2.5.1.3. Education level
Education level was re-coded by the AIID study coordinators in both the pilot and confirmatory datasets to a five-point scale based on two older items in the AIID study that provided a wider range of values but ultimately resulted in a small number of cases at the scale extremes (see codebook for details at https://

Results for the confirmatory predictions were unchanged after excluding adolescents (see electronic supplementary material, Supplemental Analyses S2).

osf.io/3sg5e/). The final five-point single-item scale consisted of the following values: (1) not a high school graduate, (2) high school graduate, (3) some college or associate's degree, (4) bachelor's degree, and (5) graduate degree or graduate education. Participants who failed to provide a response on this measure were excluded from all analyses. This variable was $z$-transformed prior to all analyses.

### 2.5.1.4. Implicit Association Test

The IAT [109] was used to implicitly measure evaluative associations for the rich and the poor. As for other contemporaneous Project Implicit data [113,114], the IAT was presented via the Internet using Java and CGI technology. This software used the respondent's computer resources to present stimuli and record response latencies, thereby reducing noise that would be caused by variable connection speeds. As noted by Nosek *et al.* [113,114], latency recording is limited by the local system's clock rate, with error windows of 16–60 ms. However, the resulting noise is not systematic, and the IAT tends to elicit large effects that are stable due to averaging data across many trials [114].

*2.5.1.4.1. Introductory blocks and stimuli.* Participants began the IAT learning the dimension of interest: rich people versus poor people. In the first block, participants categorized words indicative of the rich and poor using the 'a' (e.g. for rich) and ';' (e.g. for poor) keys. Words representing the rich people anchor included: Wealthy, Affluent, Prosperous and Well Off. Words representing the poor people anchor included: Poor, Impoverished, Broke and Bankrupt. In the second block, participants then learned the attribute dimension using one of the three randomly assigned sets of anchor terms: good versus bad, positive versus negative, or pleasant versus unpleasant. (All analyses collapsed across the three pairs of anchor terms.) As in the first block, participants categorized word stimuli as good/positive/ pleasant or bad/negative/unpleasant using the 'a' and ';' keys, respectively. For most participants ($n_{pilot} = 157$; $n_{confirmatory} = 699$), words representing the good/positive/pleasant anchor included: Appealing, Delight, Excitement, Glee, Laughing and Splendid. For this majority of participants, words representing the bad/negative/unpleasant anchor included: Animosity, Dirty, Gross, Evil, Neglected and Rotten. Due to an apparent programming error, a small subset of participants ($n_{pilot} = 18$; $n_{confirmatory} = 68$) completed the rich–poor IAT with a different set of attribute words. Good/positive/ pleasant words included: Love, Cheer, Friend, Pleasure, Paradise and Splendid. Bad/negative/ unpleasant words included: Abuse, Grief, Poison, Sadness, Pain and Bomb. All analyses collapsed across the two attribute word sets.

*2.5.1.4.2. Dual-categorization blocks and transition block.* Next, participants completed the third block. Here, the two preceding tasks were combined such that each key represented two possible categorizations. For example, the 'a' key was assigned to both rich people and good/positive/pleasant, and the ';' key was assigned to both poor people and bad/negative/unpleasant. Respondents then categorized both kinds of word stimuli (i.e. those denoting wealth and valence), which alternated throughout this 20-trial practice block. After a brief rest, participants then completed the fourth block (i.e. the first critical block). Here, participants simply repeated the same task as in the third block, but over 40 trials. In the fifth block (i.e. transition block), participants only categorized stimuli along the wealth dimension. But this time, the keys were reversed such that 'a' was assigned to poor people and ';' was assigned to rich people. Having completed the fifth block, participants then completed a 20-trial practice block where the 'a' key was assigned to both poor people and good/positive/pleasant, and the ';' key was assigned to both rich people and bad/negative/unpleasant. Finally, participants completed a final critical block of 40 trials that was otherwise identical to the preceding 20-trial practice block. As in previous work [112,114], the ordering of the third/fourth and sixth/seventh blocks was counterbalanced across participants to minimize the impact of block order effects.

*2.5.1.4.3. Scoring.* The IAT effect is calculated using latency data from the two critical blocks and their preceding practice trials. Categorizing stimuli faster when poor people share the same key as bad attributes (and when rich people are paired with good attributes) than vice versa indicates a stronger association strength between poor people and bad (and rich people and good) relative to the opposite key mapping. In other words, this would reflect an implicit pro-rich bias. The IAT $D$ scores provided in the AIID dataset were computed according to recommended guidelines, which include some trial-level exclusions (see table 4 of [111]). Specifically, the scoring algorithm for $D$ scores in the AIID pilot and confirmatory datasets is as follows:

(1) Use all trials from blocks 3, 4, 6 and 7.
(2) Eliminate any trials with latencies exceeding 10 000 ms.

(3) Compute the mean of correct latencies for each block.
(4) Compute one pooled standard deviation for all trials in blocks 3 and 6 and another for all trials in blocks 4 and 7.
(5) Replace each error latency with its respective block's mean RT, plus 600 ms.
(6) Average the resulting values of each of the four blocks.
(7) Compute two difference scores: block 6 – block 3, block 7 – block 4.
(8) Divide each difference by its associated pooled-trials standard deviation (step 4).
(9) Average the two quotients from step 8.

In summary, the $D$ score reflects an average of two subscores: (i) average response times between the two practice blocks (i.e. blocks 3 and 6) divided by the standard deviation of all response times for both blocks, and (ii) average response times between the two critical blocks (i.e. blocks 4 and 7) divided by the standard deviation of all response times for both blocks.

Due to its similarity to Cohen's $d$, the IAT $D$ score is thought to reflect the magnitude of implicit associations [114]. The IAT $D$ score served as our key dependent measure of implicit pro-rich bias, with larger positive $D$ scores reflecting a greater positive association for the rich relative to the poor. $D$ scores were standardized into $Z$ scores prior to analysis, as were all continuous independent variables. This was done to facilitate comparisons of effect sizes across analyses and studies.

### 2.5.2. Measures for exploratory analyses

In addition to the aforementioned measures for confirmatory analysis, the AIID study includes a rich set of additional variables that may be used for exploratory *post hoc* analyses. These additional measures include standard demographic variables and explicit measures tapping into thoughts and feelings about the rich–poor dimension that was assessed by the rich–poor evaluative IAT (i.e. the dependent variable used for confirmatory analysis).

#### 2.5.2.1. Demographic items

Demographics were assessed prior to the completion of other measures in this study during the participant's initial site-wide sign-up. In addition to gender, income level and education level (i.e. the independent variables for confirmatory analysis), demographic items included participants' age, citizenship, country of residence, social class, English language fluency, race/ethnicity, occupation, political identity (liberalism versus conservatism), religion, religiosity and ZIP code.

#### 2.5.2.2. Explicit measures

In addition to the IAT, participants also completed 27–29 self-report items regarding their own and others' attitudes towards rich people and poor people. These items were pulled from a pool of 76 items, randomized with some constraints (see codebook for details at https://osf.io/3sg5e/). Self-report items were grouped by the AIID coordinators into 18 different measures. Although these measures do not form the main focus of the present report, we provide descriptive statistics and some limited exploratory analyses of these measures to better contextualize our central analyses of implicit bias on the IAT. The AIID study team counterbalanced the experiment ordering such that individual difference measures preceded the IAT for some participants. Measures 1–14 were included in descriptive and correlational analyses. Measures 1–8 were additionally included in extended parallel regressions.

*(1) Personal evaluations.* Participants separately rated the rich and the poor on one of the three closely related dimensions: positivity (e.g. 'How positive or negative do you feel towards the rich?'), warmth (e.g. 'How warm or cold do you feel towards the rich?') or likeability (e.g. How much do you like or dislike the rich?'), responding on a scale from 1 (Strongly negative, Cold, or Strongly dislike) to 10 (Strongly positive, Warm, or Strongly like). The explicit evaluations measure consists of the difference in evaluations for the rich minus the poor.[4] This measure was used in the exploratory parallel regressions as an index of the self's explicit evaluations of the rich versus poor.

---

[4]This and all other explicit measures in the AIID dataset were originally computed by the AIID study team such that higher scores indicated greater pro-poor bias. In our stage 1 registered report, we conserved these original computations. However, for our final report, we opted to reverse-code all explicit measures prior to analysis so as to make them consistent with IAT $D$ scores, for which higher scores indicate greater pro-rich bias.

*(2) Others' evaluations.* Participants completed three to four items from a set of six items asking about the degree to which others (e.g. friends, family, people in general) prefer the poor over the rich. Participants responded on a scale from −3 (Strongly prefer the rich to the poor) to 3 (Strongly prefer the poor to the rich). This measure was reverse-coded and used in exploratory parallel regressions as a measure of others' evaluations of the rich versus poor.

*(3) Cultural evaluations.* Participants indicated the extent to which society at large evaluates the rich and the poor on one of three closely related dimensions: positivity (e.g. 'How positive or negative does the average person feel towards the rich?'), warmth (e.g. 'How warm or cold is society towards the rich?') or likeability (e.g. 'How much does the culture you live in like or dislike the rich?'), responding on a scale from 1 (Strongly negative, Cold or Strongly dislike) to 10 (Strongly positive, Warm or Strongly like). The cultural evaluations measure consists of the difference in evaluations for the rich minus the poor. This measure was used in the exploratory parallel regressions as an index of the cultural evaluations of the rich versus poor.

*(4) Internal pressure.* Separately for the rich and the poor, participants completed one item from a set of four items tapping into the extent to which making positive evaluations and avoiding negative evaluations of the rich/poor is consistent with their personal values. Participants responded on a scale from 1 (Strongly disagree) to 6 (Strongly agree). The internal pressure measure consists of the difference for the rich minus for the poor. This measure was used in exploratory parallel regressions as a measure of internal pressure to adjust one's evaluations based on social status.

*(5) Others' pressure.* Separately for the rich and the poor, participants completed two items from a set of eight items asking about the extent to which they moderate their attitudes towards the rich/poor in order to gain approval or avoid condemnation from others. Participants responded on a scale from 1 (Strongly disagree) to 6 (Strongly agree). The others' pressure measure consists of the difference for the rich minus for the poor. This measure was used in exploratory parallel regressions as a measure of external cultural pressure to adjust one's evaluations based on social status.

*(6) Cultural pressure.* Separately for the rich and the poor, participants completed two items from a set of eight items asking about perceptions of the average person's motivations and experience of cultural pressure to evaluate the rich/poor positively. Participants responded on a scale from 1 (Not at all motivated or Strongly disagree) to 6 (Strongly motivated or Strongly agree). The cultural pressure measure consists of the difference for the rich minus for the poor. This measure was used in exploratory parallel regressions as a measure of external cultural pressure to adjust one's evaluations based on social status.

*(7–8) Monopolar evaluations—ambivalence.* Separately for each social class, participants responded to two monopolar items: 'Thinking of only the positive things and not the negative, how positive are the rich?'; 'Thinking of only the negative things and not the positive, how positive are the rich?' Participants responded on a scale from 1 (Not at all positive/negative) to 6 (Very positive/negative).

Ambivalence was computed separately for the rich and the poor as a weighted index of the minimum intensity of positive evaluations and negative evaluations divided by the difference in magnitude between positive and negative evaluations. Concretely, the formula was as follows: $\text{Ambivalence}_{Rich} = \text{minimum}(\text{positive}_{Rich}, \text{negative}_{Rich})/(6 + \text{maximum}(\text{positive}_{Rich}, \text{negative}_{Rich}) - \text{minimum}(\text{positive}_{Rich}, \text{negative}_{Rich}))$. The index ranges from 0 to 1 (exclusive of 0), with larger scores indicating greater ambivalence, operationalized as similarly intense positive and negative scores. This ambivalence index gives greater weight to participants with equally high positive and negative scores compared to participants with equally low positive and negative scores. As for all other differences in explicit measures in this report, the difference score for ambivalence was computed as $\text{Ambivalence}_{Rich} - \text{Ambivalence}_{Poor}$.

*(9) Relative personal preference.* This measure consisted of a single item, 'Which do you prefer, the rich or the poor?' Participants responded on a scale from −3 (Strongly prefer the rich to the poor) to 3 (Strongly prefer the poor to the rich). This item was reverse-coded prior to analysis.

*(10) Gut reactions.* This measure assessed gut reactions in separate items for the rich and the poor using a single item. For example, 'People's gut reactions about a topic can be different from their feelings after they have had time to think about it. For example, someone who is trying to quit smoking might have a very positive gut reaction, but negative actual feelings toward smoking. Rate your gut reactions and actual feelings toward the topics below: Gut reactions toward the rich'. Participants responded on a scale from 1 (Strongly negative) to 10 (Strongly positive). The gut reactions measure consists of the difference in gut reactions for the rich minus the poor.

*(11) Actual feelings.* This measure assessed actual feelings in separate items for the rich and the poor using a single item. Actual feelings items began with the same preface used for gut reactions items, but

with the prompt to report actual feelings instead of gut reactions. Participants responded on a scale from 1 (Strongly negative) to 10 (Strongly positive). The actual feelings measure consists of the difference in actual feelings for the rich minus the poor.

(12) *Polarity.* In two separate items, participants indicated the perceived consequences of liking the rich for liking the poor, and vice versa. For example, 'Having positive feelings toward the rich implies having negative feelings toward the poor'. Participants responded on a scale from 1 (Strongly disagree) to 6 (Strongly agree). The polarity measure consists of the average polarity score for both items.

(13) *Identity.* Separately for the rich and the poor, participants indicated the extent to which they included these identities in their self-concept. Participants responded on a scale from 1 (None at all) to 6 (Very much). The identity measure consists of the difference in identification with the rich minus identification with the poor.

(14) *Self-concept centrality.* Separately for the rich and the poor, participants completed one item from a set of four items tapping into the extent to which accepting the rich/poor is an important part of their self-concept. Participants responded on a scale from 1 (Strongly disagree) to 6 (Strongly agree). The self-concept measure consists of the difference for the rich minus for the poor. This measure was used in exploratory parallel regressions.

(15) *Attitudinal certainty.* These items asked about participants' certainty about their feelings towards the rich and the poor. We did not analyse this measure.

(16) *Personal importance.* These items asked about the personal importance that participants placed on their feelings for the rich and the poor. We did not analyse this measure.

(17) *Affective forecasting.* These items asked about participants' expectations that their attitudes towards the rich and the poor might change over time. We did not analyse this measure.

(18) *Mindfulness.* These items asked participants about how much they think about their feelings towards the rich and the poor. We did not analyse this measure.

## 2.6. Protocol

Upon agreeing to participate in the study, AIID participants were randomly assigned to 1 of 95 different attitude domains. Of interest to the present analyses is the rich–poor domain (for other domains, see the full AIID study overview at https://osf.io/atymr/). Three out of every four participants in the rich–poor domain completed an evaluative IAT that assessed associations between this domain and evaluative anchors (e.g. good, bad). The remaining participants in the rich–poor domain instead completed an identity IAT (anchors: self, other). Our analyses focus only on participants who completed the evaluative IAT. The order of the rich–poor IAT and the explicit measures on the same dimension (see above) was randomized across participants, such that approximately half of the participants completed the IAT first and the remainder completed the explicit measures first. Finally, participants completed 20 individual difference measures. These measures are not relevant to the present analyses. Because they were assessed after all other measures, we do not list them here, but they are available at https://osf.io/atymr/.

## 2.7. Power estimates based on analyses of pilot data

In this section, we first report on the results from the pilot analyses. After reporting the pilot results, we then provide power estimates via simulation [115] for each hypothesis using parameters from our pilot analyses and our projected sample for confirmatory analyses. We report the results of our confirmatory analyses in the Results section.

### 2.7.1. Results from pilot analyses

Linear regressions were computed in R (for analysis scripts, see https://osf.io/jcgyn/) to test for the effects of participant gender, income and education on IAT $D$ scores. For the pilot data, we conducted an omnibus regression model examining the effects of gender, income, education and all possible interactions between these factors. We then followed up on this model, focusing on the Gender × Income interaction at low and high education levels (±1.5 s.d.).

#### 2.7.1.1. Relationships between independent variables

The pilot data provided no evidence of a difference between men and women in terms of income or education, $|t| < 0.62$, $p > 0.54$. However, we observed a significant correlation between our

**Figure 1.** Fitted estimates for pro-wealthy implicit bias as a function of the participant's gender and income in the pilot data. Error bars represent standard error for each estimate. Analyses revealed a significant interaction such that men (purple line) but not women (blue line) showed greater implicit pro-wealthy bias with increasing income. This two-way interaction did not replicate in the confirmatory sample. All significant simple effects are indicated with asterisks, $p < 0.05$.

standardized predictors for income and education $r_{173} = 0.22$, $p = 0.004$. Because the variance inflation factors (VIF) for all model terms in the omnibus model were below 1.5, our pilot data were not sufficiently impacted by multi-collinearity to warrant orthogonalization of income and education. This was also the case for our confirmatory dataset (see Results). Although now moot, our pre-registered plan for dealing with multi-collinearity in our confirmatory dataset (i.e. if any VIF > 10) was to orthogonalize the terms contributing to variance inflation (see https://osf.io/d5s23/). Because they tend to correlate, we anticipated this being most likely for income and education. Were this the case, we planned to orthogonalize income and education in all analyses by extracting the residuals for income in a model predicting education as a function of income. The residual variance for income, independent of its relationship with education, would have been used in all subsequent analyses.

### 2.7.1.2. H1. Men (versus women) will show greater pro-wealthy bias with increasing income
Evidence constituting support for this hypothesis would be a significant Gender × Income interaction in the omnibus model (i.e. at the sample's mean education level). In our pilot analysis, this interaction was significant (figure 1), $b = 0.132$, s.e. = 0.055, $CI_{95\%} = [0.024, 0.240]$, $t_{167} = 2.413$, $p = 0.017$. Tests of simple effects indicated that men showed increasing pro-wealthy implicit bias with increasing income levels (H1A), $b = 0.106$, s.e. = 0.044, $CI_{95\%} = [0.019, 0.193]$, $t_{167} = 2.402$, $p = 0.017$. This effect of income was non-significant for women (H1B), $b = -0.026$, s.e. = 0.032, $CI_{95\%} = [-0.090, 0.038]$, $t_{167} = -0.804$, $p = 0.423$. Additional simple effects revealed that high-income men showed greater implicit pro-wealthy bias than high-income women, $b = 0.282$, s.e. = 0.101, $CI_{95\%} = [0.083, 0.482]$, $t_{167} = 2.794$, $p = 0.006$. This simple effect of gender was non-significant at mean income, $b = 0.084$, s.e. = 0.055, $CI_{95\%} = [-0.025, 0.193]$, $t_{167} = 1.523$, $p = 0.130$, and appeared to reverse at low income, $b = -0.114$, s.e. = 0.097, $CI_{95\%} = [-0.305, 0.077]$, $t_{167} = -1.176$, $p = 0.241$, but this reversal was non-significant.

### 2.7.1.3. H2. The Gender × Income × Education interaction
In our pilot data, the Gender × Income × Education interaction was non-significant, $b = -0.089$, s.e. = 0.059, $CI_{95\%} = [-0.205, 0.027]$, $t_{167} = -1.521$, $p = 0.130$. Although this three-way interaction was non-significant, we nonetheless decomposed the interaction for the purpose of estimating the power of our follow-up models (see H2A and H2B). We focus on theoretically motivated simple effects in the text, but all significant simple effects are indicated in figure 2. To facilitate interpretation of these exploratory decompositions, we note that the main effect of education level in the omnibus model for the pilot data was significant, with greater education being associated with greater implicit pro-wealthy bias, $b = 0.111$, s.e. = 0.029, $CI_{95\%} = [0.055, 0.168]$, $t_{167} = 3.897$, $p < 0.001$.

*2.7.1.3.1. H2A and H2B. At low-income levels, men (versus women) will show greater pro-wealthy bias with increasing education.* Initial evidence constituting support for this hypothesis would be a significant Gender × Education interaction in our first follow-up model (i.e. at 1.5 s.d. below the sample's mean income level). In our pilot analysis, this interaction was non-significant (figure 2), $b =$

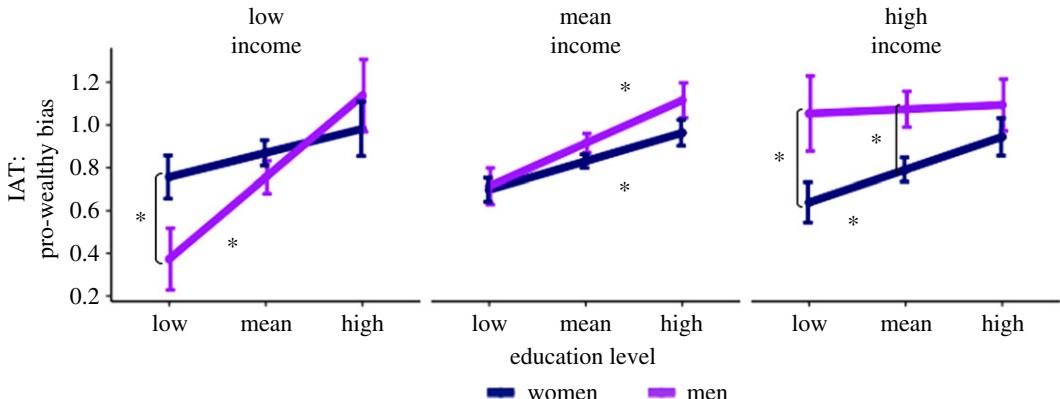

**Figure 2.** Fitted estimates for pro-wealthy implicit bias as a function of the participant's gender, education and income in the pilot data. Error bars represent standard error for each estimate. This three-way interaction did not replicate in the confirmatory dataset. All significant simple effects are indicated with asterisks, $p < 0.05$.

0.180, s.e. = 0.113, $CI_{95\%}$ = [−0.043, 0.403], $t_{167}$ = 1.591, $p$ = 0.113. We nonetheless followed up on this non-significant interaction with specific theoretically informed contrasts.

The gender roles and cultural fit accounts specifically predict that greater increasing education will lead to an attenuation of pro-wealthy bias among low-income women who face unique challenges in advancing their status through educational prestige (H2A). The strongest support for this prediction would take the form of a negative slope of education for low-income women. However, the simple effect of education on implicit pro-rich bias in low-income women trended in the opposite direction, although non-significant, $b$ = 0.075, s.e. = 0.066, $CI_{95\%}$ = [−0.054, 0.205], $t_{167}$ = 1.145, $p$ = 0.254. Nonetheless, given that increasing education led to a general increase in implicit pro-rich bias in the overall sample, it is noteworthy that low-income women (and high-income men—discussed below) were the only exceptions to this pattern. We consider this partial support for the prediction that increasing education may attenuate implicit pro-rich bias among low-income women.

Not exclusive of the gender roles and cultural fit accounts, the precarious manhood account predicts that increasing education will have the opposite effect among low-income men, who may view education as a viable means of addressing chronic status insecurity in lieu of income (H2B). The strongest support for this prediction would take the form of a positive slope of education for low-income men. Indeed, we observed that low-income men showed a significant increase in implicit pro-rich bias as a function of education, $b$ = 0.255, s.e. = 0.092, $CI_{95\%}$ = [0.073, 0.436], $t_{167}$ = 2.771, $p$ = 0.006.

*2.7.1.3.2. H2C and H2D. At high-income levels, increasing education will further enhance implicit pro-wealthy bias.* Evidence constituting support for this hypothesis would be a significant main effect of education in our second follow-up model (i.e. at 1.5 s.d. above the sample's mean income level). Although we do not have strong *a priori* predictions of a gender difference among high-income individuals, this predicted main effect may be subsumed by a Gender × Education interaction in the event that education increases implicit pro-wealthy bias to different degrees for women (H2C) and men (H2D).

In our pilot analysis, the effects of education and the Gender × Education interaction were both non-significant, $b$ = 0.058, s.e. = 0.048, $CI_{95\%}$ = [−0.038, 0.153], $t_{167}$ = 1.193, $p$ = 0.235, and $b$ = −0.089, s.e. = 0.097, $CI_{95\%}$ = [−0.279, 0.102], $t_{167}$ = −0.917, $p$ = 0.360, respectively. To parallel the contrasts reported above for low-income individuals, we tested for effects of education separately for men and women (figure 2). We found that increasing education increased implicit pro-rich bias for women, $b$ = 0.102, s.e. = 0.048, $CI_{95\%}$ = [0.008, 0.196], $t_{167}$ = 2.136, $p$ = 0.034, but not for men, $b$ = 0.013, s.e. = 0.084, $CI_{95\%}$ = [−0.153, 0.179], $t_{167}$ = 0.158, $p$ = 0.874.

### 2.7.2. Power estimates

To estimate power for all predicted effects, we ran simulations on our pilot sample in R [116] based on scripts that we adapted from Lane & Hennes [115]. In summary, we generated 1000 simulated participant-level datasets based on the sample characteristics (e.g. gender ratio, means and standard deviations for income and education) as well as the $\beta$s and standard error from the omnibus model used on the pilot data. These simulated datasets included a correlation of 0.22 between income and education. At stage 1 (projected power), each simulated dataset contained 1000 participants, which was

well below our originally anticipated final sample size. We opted for a smaller sample size to provide a more conservative projected estimate of power. In addition to our main simulations of the pilot data, we also computed parallel sensitivity analyses to provide a sense of how small the Gender × Income × Education interaction could be before falling below 80% power. All other parameters in these parallel simulations were the same as in the main simulations of the pilot data. Lastly, we also report power estimates and sensitivity analyses based on the final sample size of the confirmatory dataset (stage 2).

For each simulated dataset, we ran the omnibus model and follow-up models (see above) to test our predicted effects. Power for each test is defined as the proportion of simulations that resulted in a significant effect for that test. Analysis scripts for these simulations are available on the Open Science Framework (https://osf.io/ucf8x/).

### 2.7.2.1. Projected power at stage 1

For H1, our projected power estimate for the Gender × Income interaction was 100%. For men (H1A), the simple effect of income was powered at 100%. For women (H1B), the simple effect of income was powered at 48.3%.

For H2, our projected power estimate for the Gender × Income × Education interaction was 98.1%. Because pilot samples may overestimate the magnitude of effects [117], particularly for higher order interactions [118], we conducted additional sensitivity simulations for this three-way interaction in order to determine the smallest detectable effect size that could be detected 80% of the time. With all other effect size parameters being equal, these additional simulations indicated that a conservative sample size of 1000 participants would afford sufficient power (i.e. 80.5%) to detect a significant three-way interaction that is as much as 30% smaller than the effect size observed in our pilot sample.

Our projected power estimate for the Gender × Education interaction at 1.5 s.d. below the sample's mean income level was 99.1%. For low-income women (H2A), the simple effect of education was powered at 88.9%. For low-income men (H2B), the simple effect of education was powered at 100%.

Our projected power estimate for the main effect of education at 1.5 s.d. above the sample's mean income level was 80.1%. Our projected power estimate for the Gender × Education interaction for high-income participants was 57.7%. For high-income women (H2C), the simple effect of education was powered at 98.4%. For high-income men (H2D), the simple effect of education was powered at 5.7%.

### 2.7.2.2. Observed power at stage 2

All parameters (e.g. $\beta$s and error variance) for stage-2 sensitivity analyses remained the same as at stage 1 except that the sample size was reduced from the projected estimate of 1000 participants to final observed sample of 767 participants. For H1, our observed power estimate for the Gender × Income interaction was 100%. For men (H1A), the simple effect of income was powered at 100%. For women (H1B), the simple effect of income was powered at 40.6%.

For H2, our observed power estimate for the Gender × Income × Education interaction was 93.3%. As we did at stage 1, we conducted additional sensitivity simulations for this three-way interaction in order to determine the smallest detectable effect size that could be detected 80% of the time. With all other effect size parameters being equal, these additional simulations indicated that the final sample of 767 participants would afford sufficient power (i.e. 81.8%) to detect a significant three-way interaction that is as much as 20% smaller than the effect size observed in our pilot sample.

Our observed power estimate for the Gender × Education interaction at 1.5 s.d. below the sample's mean income level was 96.4%. For low-income women (H2A), the simple effect of education was powered at 79.6%. For low-income men (H2B), the simple effect of education was powered at 100%.

Our projected power estimate for the main effect of education at 1.5 s.d. above the sample's mean income level was 68.4%. Our projected power estimate for the Gender × Education interaction for high-income participants was 47.0%. For high-income women (H2C), the simple effect of education was powered at 96.9%. For high-income men (H2D), the simple effect of education was powered at 5.6%.

## 2.8. Contextual analyses of individual differences

Although we did not have control over the experimental design for the AIID dataset, the dataset contains many explicit measures of personal and perceived cultural attitudes towards the rich and the poor. These measures can help provide greater context for the sample's implicit pro-wealthy bias and also allow for comparisons between self-report and implicit measures of pro-wealthy bias. In this section, we summarize results from these measures in both the pilot and confirmatory datasets.

### 2.8.1. Sample descriptives

In both the pilot and confirmatory datasets, we found little evidence of perceived polarity in attitudes about the rich and poor; participants tended to disagree that evaluating the poor negatively necessarily meant having positive attitudes towards the rich (and vice versa). This is consistent with findings from Horwitz & Dovidio [9] showing that pro-rich attitudes were not predicted by anti-poor attitudes, and vice versa. With respect to individuals' attitudes, the pilot data revealed a pro-rich bias for virtually all measures tapping into personal beliefs about the rich and the poor, consistent with findings from a more recent dataset [13]. This pattern was mostly consistent with results from the confirmatory dataset. However, two exceptions emerged here. In the confirmatory data, we found that relative personal preference (i.e. extent of preference for the rich versus poor, or vice versa) and initial gut reactions favoured the poor over the rich. These two effects were non-significant in the pilot data (table 3). In sum, participants appeared to favour the poor in their immediate comparisons or intuitive responses, but they favoured the rich in their more deliberative evaluations about each group independent of the other.

Interestingly, for measures that probed perceptions of others' and cultural attitudes towards the rich and poor, participants in the pilot dataset (i) believed that society at large generally favours the poor over the rich, and (ii) felt pressure from others to moderate their explicit attitudes about the poor more so than their attitudes about the rich. These results were again largely consistent with findings from the confirmatory dataset, with one exception. In the confirmatory dataset, we found evidence of greater cultural pressure to evaluate the rich (versus poor) in a particular fashion. This effect was non-significant in the pilot data (table 3). Taken together, these descriptive analyses show that participants generally favoured the rich in their deliberative evaluations at the individual level (cf. relative personal preference) but evidently believed their evaluations contrasted with those of society at large.

### 2.8.2. Correlations with IAT

To determine whether there were any relationships between implicit pro-rich bias and explicit measures of attitudes about the poor and rich, we conducted correlations between the IAT and difference scores on a host of explicit measures (table 4). In both the pilot and confirmatory datasets, these correlations revealed that greater pro-rich bias on the IAT was predicted by *less* explicit pro-rich bias and internal pressure to upregulate pro-rich bias. This was mostly the case for explicit measures tapping into individual attitudes (or self-concept centrality—see confirmatory data in table 4) rather than perceptions about others' or cultural attitudes. The only exception to this general trend was observed in the confirmatory dataset and involved participants' perceptions of cultural pressure about how to evaluate the rich (versus poor). The greater the perceived cultural pressure to evaluate the rich in a prescribed fashion (irrespective of valence), the greater the participant's implicit pro-wealthy bias (table 4). For further elaboration on these exploratory findings, see the Discussion section.

### 2.8.3. Parallel regressions in the pilot dataset

At the encouragement of an expert reviewer, we also conducted additional parallel regressions meant to predict scores on a selection of the aforementioned difference scores on several self-report items. Where possible, we focused on multi-item rather than single-item measures of attitudes towards the rich and the poor (see exploratory measures for descriptions). Separately for the self, generic 'others' and society at large, we analysed (i) differences in explicit evaluations (rich minus poor) and (ii) differences in felt pressure to adjust one's evaluations (rich minus poor). This resulted in a total of six parallel models. We also included two additional models examining: (i) the difference in the degree to which accepting the rich (versus poor) is important to the participant's self-concept, and (ii) the difference in ambivalence felt towards the rich (versus poor).

#### 2.8.3.1. Stage 1 summary of parallel regressions in the pilot dataset

In our stage 1 pilot dataset, the parallel regression analyses of individual differences revealed relatively few effects (see electronic supplementary material, Supplemental Analyses S1). Among the few effects we observed, it would seem that increasing income/education was generally associated with diminished explicit pro-rich bias, but only for composite measures tapping into participants' own explicit evaluations of the poor and rich. Although our sample for pilot analyses of attitudinal ambivalence was relatively small, we observed here a potentially important difference with respect to income and education. Whereas increasing education was associated with greater ambivalence towards the rich,

**Table 3.** Descriptive statistics from pilot data for select individual difference measures. Except for overall polarity, all *t*-tests are one-sample *t*-tests against zero (i.e. no difference between the rich and the poor). Bold text highlights significant effects, *p* < 0.05.

| dataset | measures (rich–poor) | M | CI95% | t | d.f. | p-value |
|---|---|---|---|---|---|---|
| pilot data | 1. personal evaluations | 0.706 | [0.333, 1.078] | 3.736 | 162 | **<0.001** |
| | 2. others' evaluations | −1.022 | [−1.174, −0.870] | −13.246 | 158 | **<0.001** |
| | 3. cultural evaluations | −2.988 | [−3.502, −2.474] | −11.475 | 162 | **<0.001** |
| | 4. internal pressure | 0.691 | [0.430, 0.953] | 5.214 | 161 | **<0.001** |
| | 5. others' pressure | −0.141 | [−0.282, −0.000] | −1.975 | 162 | **0.050** |
| | 6. cultural pressure | 0.031 | [−0.145, 0.206] | 0.346 | 162 | 0.730 |
| | 7/8. ambivalence | 0.034 | [−0.031, 0.099] | 1.033 | 63 | 0.306 |
| | 9. relative personal preference | −0.019 | [−0.236, 0.198] | −0.171 | 159 | 0.865 |
| | 10. gut reactions | 0.327 | [−0.124, 0.778] | 1.434 | 164 | 0.154 |
| | 11. actual feelings | 0.594 | [0.187, 1.001] | 2.881 | 164 | **0.004** |
| | 12. overall polarity (rich and poor)[a] | 2.680 | [2.290, 3.071] | −4.198 | 60 | **<0.001** |
| | 13. identity | 0.051 | [−0.364, 0.467] | 0.246 | 77 | 0.807 |
| | 14. self-concept centrality | 0.945 | [0.691, 1.199] | 7.348 | 162 | **<0.001** |
| confirmatory data | 1. personal evaluations | 0.318 | [0.128, 0.509] | 3.281 | 722 | **0.001** |
| | 2. others' evaluations | −1.068 | [−1.139, −0.997] | −29.704 | 723 | **<0.001** |
| | 3. cultural evaluations | −3.156 | [−3.396, −2.917] | −25.842 | 722 | **<0.001** |
| | 4. internalpressure | 0.624 | [0.530, 0.719] | 12.986 | 726 | **<0.001** |
| | 5. others' pressure | −7.464 | [−0.354, −0.207] | −7.464 | 721 | **<0.001** |
| | 6. cultural pressure | 0.092 | [0.010, 0.174] | 2.206 | 726 | **0.028** |
| | 7/8. ambivalence | 0.016 | [−0.014, 0.047] | 1.065 | 289 | 0.288 |
| | 9. relative personal preference | −0.256 | [−0.352, −0.159] | −5.203 | 718 | **<0.001** |
| | 10. gut reactions | −0.267 | [−0.490, −0.045] | −2.360 | 728 | **0.019** |
| | 11. actual feelings | 0.235 | [0.052, 0.417] | 2.526 | 727 | **0.012** |
| | 12. overall polarity (rich and poor)[a] | 2.462 | [2.311, 2.612] | −13.555 | 299 | **<0.001** |
| | 13. identity | −0.099 | [−0.295, 0.097] | −0.996 | 322 | 0.320 |
| | 14. self-concept centrality | 0.588 | [0.474, 0.701] | 10.166 | 724 | **<0.001** |

[a]One-sample *t*-test against 3.5 (neither agree nor disagree).

increasing income was associated with reduced ambivalence towards the rich among male participants. In contrast with the findings for measures of individual attitudes, we observed no effects of external pressure to modulate class-based attitudes (cf. personal versus others' and cultural attitudes). With only the pilot data in hand, our best interpretation of these results was that individuals with higher status (viz., through some combination of income, education and/or gender) tend to downregulate their positive evaluations of the rich, perhaps out of greater ambivalence among the educated and/or a desire to minimize apparent privilege [119] among the wealthy. Although we do not have the means to test the precise mechanism for status-based downregulation of pro-rich bias in this dataset, our intention was to follow up on these analyses in the main confirmatory dataset, providing the

**Table 4.** Correlations with IAT D scores (Pro-Rich Bias). Polarity reflects average perceived polarity for rich and poor. Bold text highlights significant effects, $p < 0.05$.

| measures (rich–poor) | pilot data | | | confirmatory data | | |
|---|---|---|---|---|---|---|
| | $r$ | $n$ | $p$-value | $r$ | $n$ | $p$-value |
| 1. personal evaluations | **−0.26** | **163** | **0.001** | **−0.22** | **723** | **<0.001** |
| 2. others' evaluations | **−0.16** | **159** | **0.039** | **−0.15** | **724** | **<0.001** |
| 3. cultural evaluations | −0.002 | 163 | 0.798 | 0.05 | 723 | 0.178 |
| 4. internal pressure | **−0.16** | **162** | **0.036** | **−0.10** | **727** | **0.005** |
| 5. others' pressure | 0.06 | 163 | 0.436 | 0.01 | 722 | 0.703 |
| 6. cultural pressure | −0.03 | 163 | 0.683 | **0.08** | **727** | **0.024** |
| 7/8. ambivalence | −0.10 | 64 | 0.440 | −0.07 | 290 | 0.224 |
| 9. relative personal preference | **−0.19** | **160** | **0.017** | **−0.26** | **719** | **<0.001** |
| 10. gut reactions | **−0.22** | **165** | **0.004** | **−0.22** | **729** | **<0.001** |
| 11. actual feelings | **−0.24** | **165** | **0.002** | **−0.21** | **728** | **<0.001** |
| 12. overall polarity (rich and poor) | −0.02 | 61 | 0.897 | −0.06 | 300 | 0.273 |
| 13. identity | **−0.24** | **78** | **0.033** | **−0.19** | **323** | **0.001** |
| 14. self-concept centrality | −0.11 | 163 | 0.175 | **−0.10** | **725** | **0.007** |

reader with (i) a replication of these piloted findings, (ii) an extended discussion of how they may shed further light on the implicit/explicit divergence in pro-rich biases that occurs for high-status participants, and (iii) additional discussion of future directions that may help uncover the mechanism underlying status-based downregulation of pro-rich bias.

### 2.8.3.2. Stage 2 summary of parallel regressions in the confirmatory dataset

With the exception of the main effect of gender in the analysis of self-concept centrality, exploratory analyses of self-reported beliefs and attitudes in the confirmatory dataset failed to support any of the preliminary findings from our stage 1 analysis of the pilot dataset (see electronic supplementary material, Supplemental Analyses S1). Nonetheless, in the larger samples afforded by the confirmatory dataset, a pattern of significant findings emerged that merits discussion. Namely, across all measures implicating individual-level beliefs and attitudes (viz., personal evaluations, internal pressure and self-concept centrality), we observed interactions between participant gender, income and education. Overall, men who showed the most pro-rich bias were congruent in their income and education levels (i.e. either low or high in both income and education). Men who showed the least pro-rich bias were incongruent in their income and education levels (e.g. low income but high education). Women who showed the most pro-rich bias were low in income but high in education. Women who showed the least pro-rich bias were generally high in income—although this varied depending on the measure of explicit bias. In the discussion, we will return to these findings in the context of the main results presented in the following section.

## 3. Results

### 3.1. Relationships between independent variables

The confirmatory dataset provided no evidence of a difference between men and women in terms of income or education, $|t| < 0.57$, $p > 0.29$. However, we observed a significant correlation between our standardized predictors for income and education $r_{766} = 0.14$, $p < 0.001$. Because the VIF for all model terms in the omnibus model were below 1.5, our confirmatory dataset was not sufficiently impacted by multi-collinearity to warrant orthogonalization of income and education as outlined in our pre-registered analysis plan.

## 3.2. Confirmatory analysis

In our analysis of IAT $D$ scores as a function of gender, income, education and all possible interactions between these predictors, the only significant effect was the main effect of gender, $b = 0.065$, s.e. = 0.028, $CI_{95\%} = [0.009, 0.121]$, $t_{759} = 2.292$, $p = 0.022$. Men showed greater implicit pro-rich bias than did women. Although the main effect of gender was directionally consistent with the main effect of gender in the pilot dataset (especially at average or high-income levels), we expected the effect of gender to be moderated by self-reported education and income. All other effects were non-significant, $p > 0.057$. We now turn to the results from our primary pre-registered predictions.

### 3.2.1. H1: Gender × Income interaction

The Gender × Income interaction was non-significant, $b = -0.027$, s.e. = 0.029, $CI_{95\%} = [-0.084, 0.029]$, $t_{759} = -0.956$, $p = 0.339$. Despite the absence of a formal interaction, we nonetheless followed up on our pre-registered follow-up analyses for men (H1A) and women (H1B). Contrary to expectations, income increased implicit pro-rich bias for women, $b = 0.038$, s.e. = 0.017, $CI_{95\%} = [0.005, 0.072]$, $t_{759} = 2.279$, $p = 0.023$, but not for men, $b = 0.011$, s.e. = 0.023, $CI_{95\%} = [-0.034, 0.056]$, $t_{759} = 0.480$, $p = 0.632$.

### 3.2.2. H2: Gender × Income × Education interaction

The predicted three-way interaction was also non-significant, $b = 0.007$, s.e. = 0.027, $CI_{95\%} = [-0.045, 0.060]$, $t_{759} = 0.272$, $p = 0.786$. Despite the absence of a formal interaction, we again conducted our pre-registered follow-up analyses testing for effects of education level separately for four different participant groups. Effects of education on implicit pro-rich bias were non-significant for all: low-income women (H2A), $b = -0.031$, s.e. = 0.031, $CI_{95\%} = [-0.093, 0.030]$, $t_{759} = -1.009$, $p = 0.314$, low-income men (H2B), $b = 0.011$, s.e. = 0.023, $CI_{95\%} = [-0.034, 0.056]$, $t_{759} = 0.296$, $p = 0.768$, high-income women (H2C), $b = -0.036$, s.e. = 0.029, $CI_{95\%} = [-0.093, 0.022]$, $t_{759} = -1.222$, $p = 0.222$ and high-income men (H2D), $b = 0.029$, s.e. = 0.040, $CI_{95\%} = [-0.049, 0.107]$, $t_{759} = 0.721$, $p = 0.471$.

# 4. Discussion

## 4.1. Confirmatory analysis

Results from our confirmatory analyses contribute to a growing body of work illustrating that people and cues of high status elicit more positive implicit associations [9,14,15,17]. Our pre-registered analyses of the IAT data build on this literature by showing greater pro-rich/anti-poor implicit bias in men than in women. This finding is complemented by a recent neuroimaging study showing evidence of greater activity in brain regions indicative of positive evaluations when men (versus women) formed impressions of high-status individuals [120]. Implicit and neural preferences for the wealthy among men is consistent with gender roles, which associate men more than women with positions of higher status or prestige [20,33–36], and with masculine gender identity, which is predicated partly on status attainment and maintenance [53–55].

Importantly, our central predictions that women's and men's implicit pro-rich bias would be modulated by income and education were unsupported. One potential explanation may be that the sample was underpowered to detect these effects. However, a cursory inspection of effect sizes suggests that the standardized $\beta$ coefficients for H1 and H2 were much smaller in our complete dataset than in the pilot dataset. Although our pilot dataset approached 50 participants per cell for H1, conforming to recommendations for exploring unknown effects [121], even pilot samples of this size can overestimate effect sizes in pilot data, particularly if true effects are small to non-existent [117]. Contrary to plausible predictions generated from the literature on gender roles and masculine gender identity, we conclude that implicit pro-wealth bias among men is robust and less likely than expected to be shaped by socioeconomic rank (e.g. income, education).

## 4.2. Exploratory analyses of self-report measures

Although we found no evidence that income and education modulated the effect of gender on implicit pro-rich bias, exploratory analyses of our confirmatory dataset revealed that these predictors did collectively shape explicit attitudes and beliefs about the poor. These effects were found primarily for

measures of individual attitudes and beliefs rather than perceptions of others' or cultural attitudes and beliefs. We first summarize how gender shaped explicit attitudes and beliefs before reviewing how gender effects were modulated by income and education levels.

Contrasting with our finding that men show greater implicit pro-wealthy bias, women showed greater pro-wealthy bias than men on explicit measures. Women indicated more than men that: (i) they view rich people more positively than poor people, (ii) making positive evaluations of the rich was more acceptable due to their personal values than making positive evaluations of the poor, and (iii) accepting the rich was more central to their self-concept than accepting the poor. However, in contrast with the implicit bias findings (discussed in more detail in the following section), these effects of gender on explicit beliefs and attitudes varied as a function of the participants' income and education levels. We summarize and discuss the exploratory findings from analyses of self-report data separately for men and women.

### 4.2.1. Effects of income and education on men's explicit pro-wealthy bias

For men, congruence of income and education levels contributed to the highest levels of explicit pro-rich bias. Although it might make sense that unambiguously high-status men evaluated their ingroup (i.e. rich people) more positively than less privileged men, it is less clear why this was also the case for unambiguously low-status men compared to men who arguably have some claim to high status (e.g. due to high income or high education levels). As for unambiguously low-status men, it is possible that they view the rich positively due to popular beliefs about upward mobility [122] and general acceptance of the social order [123,124]. Among Americans, beliefs about mobility are popular—even if unwarranted [122,125–127]—perhaps especially in times of economic prosperity that characterized the timeframe for data collection (just prior to the Great Recession of 2008). For low-status men, the rich could represent a kind of masculine ideal they hope to someday achieve [53,55]. As for men who are high in income or education (but not both) and who rated the rich less favourably relative to the poor, one intriguing possibility here is that these ambiguously high-status men (e.g. due to low education or income, respectively) may hold more ambivalent feelings about high-status people in general. For example, this may come from the experiencing downward social comparisons that may conflict with where individuals feel they stand in a given social hierarchy (see [23,95,128]) and a perceived lack of fit in certain elite institutions (e.g. universities: [89,93,94,97]). Future research could shed more light on this question by using a measure that directly measures ambivalence towards the rich and the poor.[5]

### 4.2.2. Effects of income and education on women's explicit pro-rich bias

For women, lower income irrespective of education predicted higher pro-rich bias in explicit evaluations of the rich versus poor (electronic supplementary material, figure S3, Supplemental Analyses S1). As women's income levels increased, their explicit pro-rich bias was attenuated. This pattern was more complex when women responded about how evaluating the rich (versus poor) positively was consistent with their personal values and self-concept. Again, low-income women showed the greatest pro-rich bias, but this was driven by low-income women who were highly educated (see electronic supplementary material, figures S4 and S5, Supplemental Analyses S1). This finding ran counter to our predictions for IAT data in hypothesis H2A that were motivated by the cultural fit account. This account argues that low-status women's evaluations of high-status people may be tempered through perceived lack of fit, particularly in male-dominated areas of higher education [129,130]. One possible interpretation of the present findings from the mate selection literature may be that women value high status in prospective mates, particularly when low on resources (cf. [131,132]) or living in less egalitarian countries [133]. However, it is not clear why this may be especially the case for low-income women who are high in education compared to those who are low in education. Another possibility worth exploring in future research may be that women from low-income backgrounds who graduate from college develop positive beliefs about the rich through their own experience of and hopes for upward mobility [134].

[5]The present study computed an indirect measure of attitudinal ambivalence towards the rich and the poor, but results were inconclusive. Part of the reason for this may have been that these analyses of ambivalence were underpowered—the final sample for the ambivalence measure was approximately one-third of the total sample size.

## 4.3. Convergences and divergences between implicit and explicit pro-rich bias

As mentioned in the previous section, it is worth noting that gender effects on implicit and explicit measures were in opposite directions. Whereas men showed greater pro-wealthy bias on the IAT, women showed greater pro-wealthy bias on explicit self-report measures, depending on their income and education levels (see electronic supplementary material, figures S3–S5, Supplemental Analyses S1). To explore the relationship between implicit and explicit measures of pro-rich bias in greater detail, we conducted correlations independent of perceiver characteristics. Overall, correlations confirmed that greater pro-wealthy bias on the IAT was predicted by *less* explicit pro-wealthy evaluative bias on all explicit measures showing a gender difference (personal evaluations of the rich, personal value of upregulating evaluations of the rich and self-concept centrality of evaluating the rich positively, among others: table 4). The only exception to this general trend was observed in the confirmatory dataset and involved participants' perceptions of cultural pressure about how to evaluate the rich (versus poor). However, this cultural pressure variable did not specify the evaluative direction of cultural pressure. In other words, greater perceived cultural pressure to evaluate the rich in a prescribed fashion (irrespective of valence) predicted greater implicit pro-wealthy bias.

Although these correlations may seem puzzling at first glance, we are not the first to find divergent patterns of evaluation based on wealth using implicit versus explicit measures. Previous research using the IAT has found that poor [17] and middle-class [9] participants preferred their respective ingroups relative to the rich on explicit measures. However, both groups showed a clear pro-rich bias on the rich–poor and rich–middle-class IATs, respectively. Using both correlational and categorical analytic approaches, Rudman *et al.* [17] observed a status-based gradient in the extent of perceivers' divergence in their implicit and explicit ingroup biases. Generally, members of powerful high-status groups (e.g. rich/White people) were consistent in preferring their own group (versus poor/Asian people) on both implicit and explicit measures (see also [123,135]). Low-power groups (e.g. the poor, minorities) showed increasingly large discrepancies on implicit and explicit measures. The least powerful group (viz., poor people) showed both a sizeable explicit ingroup bias and the strongest outgroup bias on the IAT relative to all other low-power groups. In fact, poor participants' implicit preference for the rich was comparable in magnitude to implicit ingroup biases from participants belonging to powerful high-status groups. Taken together, results from Rudman *et al.* suggest that this negative relationship between implicit and explicit measures may be due to low participant status. Exploratory analyses of our confirmatory dataset suggest a more complicated relationship that critically depends on perceiver gender and status. Unambiguously high-status or low-status men (defined in terms of congruent income and education) and high-income women tended to show the greatest convergence in their implicit and explicit attitudes, with men showing more pro-wealthy bias on the IAT and explicit measures and women showing less pro-wealthy bias on the same measures. By contrast, men with incongruent income/education (e.g. high income, low education) and low-income women (especially at higher education levels) showed the most divergence between implicit and explicit attitudes. Whereas men showed higher pro-rich bias on the IAT and lower pro-rich bias on explicit measures, women showed higher pro-rich bias on explicit measures and lower pro-rich bias on the IAT.

Given the differences between implicit and explicit measures of pro-rich bias in our findings and those of others, some discussion is warranted about potential mechanisms that underlie these complexities. Based on the MODE model [136], it is thought that these differences emerge when spontaneous group associations (assessed with the IAT) receive further elaboration as participants formulate their self-reported evaluation of a given group (e.g. the rich). The degree to which self-reported evaluations of the rich and poor are determined by initial spontaneous associations depends on whether the participants (i) are motivated to adjust their self-reported evaluations, and (ii) have the time and mental resources to do so. Given that participants in the present study (i) reported cultural norms about evaluating the poor more favourably, and (ii) were not constrained by time limits or additional cognitive load when they completed the explicit measures, it would seem that the experimental context could increase the chances of observing diverging patterns of implicit and explicit bias.

More recent work on implicit biases suggests that they may reflect culturally prevalent associations that are nonetheless highly sensitive to context [137]. As a result, aggregate IAT measures often strongly predict outcomes at aggregate levels (e.g. counties) but less reliably predict outcomes at the individual level (e.g. participants). If so, then one might expect individual differences that are tied to relatively fixed cultural differences to more strongly impact IAT scores. For example, although

scholarly work has illustrated pervasive links between status and unique cultural practices and views [87,88,93], relative status within social hierarchies can also be subjective [138,139] and contextually malleable in ways that are not typically possible for gender (e.g. [127,140–143]). Applying this theoretical lens in hindsight, the fact that labile status characteristics modulated explicit but not implicit pro-rich bias in our confirmatory dataset is perhaps unsurprising. Of course, this *post hoc* interpretation requires further investigation. We, therefore, refrain from further speculation about mechanisms underlying divergences between implicit and explicit measures of pro-wealthy bias, pending more targeted experiments. Despite the current gap in our knowledge about the origins of discrepancies in implicit and explicit measures of status bias (or any other bias), the study of implicit bias is nonetheless important; previous work has found that implicit measures can be better predictors of real-world behaviours than are explicit measures, particularly when participants are motivated and able to modulate their explicit evaluations [144,145]. For example, the rich-middle-class IAT (rather than parallel self-report measures) predicted leniency on a rich driver who was responsible for a car accident [9].

# 5. Conclusion

Gender and social status are important components of how we see ourselves [53,55] and others [10,11,21]. The findings from this registered report provide a comprehensive overview of how implicit and explicit biases favouring the rich over the poor may depend on perceiver gender, income and education. Although our main predictions that income and education would modulate gender differences in implicit pro-rich bias were unsupported, we nonetheless found preliminary evidence that (i) men showed more implicit pro-rich bias and less explicit pro-rich bias than women, and (ii) implicit and explicit measures of pro-rich bias were inversely correlated. Gender differences in explicit pro-rich biases in particular showed some sensitivity to individual differences in income and education that may be generative of future research questions on gender differences in status-based evaluations, broadly construed.

Ethics. Participants voluntarily completed the AIID study, and no inducements or incentives from the AIID research team were used. At the start of the study, participants viewed an introduction screen that included an informed consent agreement stating that participation was voluntary, non-compensated and with minimal risk. The informed consent agreement also stated that responses were confidential and anonymous, protected and analysed in the aggregate, and with information about whom to contact in the event of any queries. Also included was the Project Implicit privacy policy. Consent was presumed through participation in the study. Data were collected in accordance with University of Virginia IRB protocol no. 2003017300.

Data accessibility. All data and analysis scripts are provided on the Open Science Framework at https://osf.io/jcgyn/ [146]. This includes data for the pilot analyses and scripts from the pilot and confirmatory analyses. At present, we are unable to provide the data for the confirmatory analyses due to a signed agreement with the AIID Study Team (see https://osf.io/65stq/). However, when we are allowed to publicly post the data, we will provide it on our OSF repository for this registered report.

Authors' contributions. B.D.M. conducted all analyses, drafted the manuscript and provided major revisions; J.C. provided critical interpretation of findings and revised the manuscript. Both authors gave final approval for publication and agree to be held accountable for the work performed therein.

Competing interests. The authors declare no competing interests for this project.

Funding. The authors declare no sources of funding for this project.

Acknowledgements. The authors would like to thank Jennifer T. Kubota for feedback on the analysis plan. The authors would also like to thank the AIID team for facilitating access to the data analysed in this report. Finally, the authors thank Michael W. Kraus and an anonymous second reviewer for their invaluable insights and suggestions.

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
