## [Reviewer comments · Royal Society Open Science]

Review History

RSOS-191232.R0 (Original submission)

Review form: Reviewer 1 (Benedict Jones)

Is the language acceptable?

Yes

Do you have any ethical concerns with this paper?

No

Have you any concerns about statistical analyses in this paper?

No

Recommendation?

Accept with minor revision

Comments to the Author(s)

This is quite far outside my own area, but the rationale for the study is very clearly specified and, as far as I can tell, well grounded in the literature. The hypotheses are also clearly specified and

follow directly from previous work on the topic. Preliminary data is presented and related to them.

The analysis pipeline is generally very clear and appropriate, but several points would benefit from clarification, in my view.

P7. It wasn't initially clear to me that the rich-poor IAT was one of the 95 different topics that participants could be randomly assigned to.

P8. Would be useful to list all the variables relevant to the confirmatory and exploratory analyses, clearly labeling which are relevant to which type of analysis.

P9. I would mention here and archive the letter from Ebersole confirming you have not had access to the confirmatory portion of the data. (In general, a little more detail about the scheme through which you have accessed these data would be welcome, but probably not essential)

P9. It took me a few reads over to get what you meant by unique participants. Could clarify.

P9. Here (and in a couple of other places) you mention exclusion criteria being 'in line with...' recommendations made by others. It would be useful to be more precise. If you mean you will (or have) applied all those that they recommend, I'd list them specifically and state them more clearly.

P10. Same issue as previous point but for 'following guidelines offered...'

P9 and P10. It would be useful to give some justification for these exclusion criteria. Yes, they are being spelled out upfront (notwithstanding the points above), but a justification or explanation for each would be handy.

P11. You say it is difficult to know in advance how many participants will be in the confirmatory data set following exclusions. I'd argue it is impossible to know, so would recommend saying how many participants you will have prior to exclusion and approximately what proportion you'd expect to exclude based on your pilot analyses.

I really liked the power analysis by simulation and the assumptions made are fair (in my view).

The level of detail in the methods and analysis pipeline is sufficient to replicate the work (but see points above where some clarifications are required).

Positive controls. My only major concern about this project is the lack of positive controls or manipulation checks. It might be that there are some form of attention checks in the original Project Implicit study that I am unaware of or some additional checks that could be added to the analysis pipeline (potentially as robustness checks?). It's hard for me to suggest any here, given I am not an expert with the IAT, but anything the authors can add to increase confidence in the procedures employed would be welcome (even if it was just an outline of robust findings replicated so far in Project Implicit, for example).

Review form: Reviewer 2 (Michael Kraus)

Is the language acceptable?

Yes

Do you have any ethical concerns with this paper?

No

Have you any concerns about statistical analyses in this paper?

Yes

Recommendation?

Major revision

Comments to the Author(s)

I (Michael Kraus) think this is a really nice use of the AIID dataset. I hope my comments here are helpful as you prepare for the full analysis. Examining gender/income/education associations with pro wealth bias is an important consideration that will link up with how policy makers and the public think about social safety net policies, as well as potentially, progressive taxation policies. I also think the proposed work is reported precisely and rigorously, and indeed, the study is well positioned for a RR format because we'd all be very interested in the results no matter what is found here. My comments, that I hope will be used to improve the manuscript, are below:

INTRO

I wonder if it makes sense to frame the gender hypotheses here more in terms of privilege v. masculinity. Or instead, it'd be better to talk about how both matter in the context of this research. Privilege matters because men have favored status in most societies and so their preference for wealth might be about a failure to acknowledge privilege or to deny advantage. However, male gender roles also would suggest a link between wealth and masculinity and so this pro-wealth bias may be less about privilege and more about masculinity. Right now, the privilege angle comes through but the gender role angle does not. I'm thinking of the precarious manhood work of vandello, the benevolent sexism work of Glick, Vescio, and Alice Eagly's considerable scholarship on this topic.

I'm with you on the hypothesis for the gender x income interaction but I get lost on the education piece in part because education is a multifaceted construct. With respect to wealth, some people learn how to accumulate wealth, so there is a learning component. There is also a status component as in, I'm yalie! There is also a culture component, being socialized around higher status people. As it stands I'm not sure which is operating in the 3-way interaction, and the argument here currently isn't precise enough about what education is and how it might relate to pro-wealth bias as a function of gender and income. All of this probably needs to be more precise so that the prediction becomes clearer. In fact, thinking on the different dimensions of education might lend itself to competing predictions!

ANALYSIS PLAN

I'm a little worried about statistical power for the three way interaction. I know the sims suggest that there is power, but I believe statistical power for interactions rests almost entirely on the shape of the interactions (data colada piece on now way interactions)--so if the pilot data is an overestimate of the true effect in the larger AIID data, you'll be under powered to test this three way interaction hypothesis. I'm not sure there is much you can do about this challenge in these data, but I do believe that the current power analyses are an overestimate, and you might consider simulating a range of power analyses to better get a sense of the uncertainty there. So if the shape is less extreme for the three-way interaction, what power do you have?

I am also wondering if the authors would consider analyzing some of the self-report data in their analysis plan. We have pro-wealth bias as DV here, but it would clarify what we find here if some of the self-report measures were also examined in a parallel regression analysis. For instance, I think there is information to be gained about understanding how much being rich/poor is consistent with the self or with felt cultural pressure. The parallel gender x income interaction with these measures might provide clues about what aspects of gender/education relate to pro-

wealth bias. Without analyzing these data we may not learn as much as we could from these data. Of course, in my own exploring of these pilot data I think there are some limits to what people can analyze on the self-report end, but it's worth considering surely.

Also things don't "approach significance" ;)

Really interesting! Best of luck!

Decision letter (RSOS-191232.R0)

15-Aug-2019

Dear Dr Mattan,

The Editors assigned to your Stage 1 Registered Report ("A registered report on how implicit pro-rich bias is shaped by the perceiver's gender and socioeconomic status") have now received comments from reviewers. We would like you to revise your paper in accordance with the referee and editors suggestions which can be found below. Please note this decision does not guarantee eventual acceptance.

When submitting your revised manuscript, you must respond to the comments made by the referees and upload a file "Response to Referees" in "Section 2 - File Upload". Please use this to document how you have responded to the comments, and the adjustments you have made. In order to expedite the processing of the revised manuscript, please be as specific as possible in your response.

Kind regards,
Professor Chris Chambers
Royal Society Open Science
openscience@royalsociety.org

Editor Comments to Author (Professor Chris Chambers):

Two reviewers have now appraised the manuscript. Both are broadly positive but nevertheless identify a number of key issues that will need to be addressed in a revision. Reviewer 1 highlights areas where greater detail and rationale are required (e.g. concerning the the exclusion criteria), and also notes the lack of outcome-neutral quality checks (e.g. positive controls). Reviewer 2 questions several aspects of the general rationale, the appropriateness of the power analysis, and the potential to report additional dimensions of the data. Overall, the issues raised appear readily addressable -- albeit substantial -- therefore a major revision is recommended.

Comments to Author:

Reviewer: 1

This is quite far outside my own area, but the rationale for the study is very clearly specified and, as far as I can tell, well grounded in the literature. The hypotheses are also clearly specified and follow directly from previous work on the topic. Preliminary data is presented and related to them.

The analysis pipeline is generally very clear and appropriate, but several points would benefit from clarification, in my view.

P7. It wasn't initially clear to me that the rich-poor IAT was one of the 95 different topics that participants could be randomly assigned to.

P8. Would be useful to list all the variables relevant to the confirmatory and exploratory analyses, clearly labeling which are relevant to which type of analysis.

P9. I would mention here and archive the letter from Ebersole confirming you have not had access to the confirmatory portion of the data. (In general, a little more detail about the scheme through which you have accessed these data would be welcome, but probably not essential)

P9. It took me a few reads over to get what you meant by unique participants. Could clarify.

P9. Here (and in a couple of other places) you mention exclusion criteria being 'in line with...' recommendations made by others. It would be useful to be more precise. If you mean you will (or have) applied all those that they recommend, I'd list them specifically and state them more clearly.

P10. Same issue as previous point but for 'following guidelines offered...'

P9 and P10. It would be useful to give some justification for these exclusion criteria. Yes, they are being spelled out upfront (notwithstanding the points above), but a justification or explanation for each would be handy.

P11. You say it is difficult to know in advance how many participants will be in the confirmatory data set following exclusions. I'd argue it is impossible to know, so would recommend saying how many participants you will have prior to exclusion and approximately what proportion you'd expect to exclude based on your pilot analyses.

I really liked the power analysis by simulation and the assumptions made are fair (in my view).

The level of detail in the methods and analysis pipeline is sufficient to replicate the work (but see points above where some clarifications are required).

Positive controls. My only major concern about this project is the lack of positive controls or manipulation checks. It might be that there are some form of attention checks in the original Project Implicit study that I am unaware of or some additional checks that could be added to the analysis pipeline (potentially as robustness checks?). It's hard for me to suggest any here, given I am not an expert with the IAT, but anything the authors can add to increase confidence in the procedures employed would be welcome (even if it was just an outline of robust findings replicated so far in Project Implicit, for example).

Reviewer: 2

Comments to the Author(s)

I (Michael Kraus) think this is a really nice use of the AIID dataset. I hope my comments here are helpful as you prepare for the full analysis. Examining gender/income/education associations with pro wealth bias is an important consideration that will link up with how policy makers and the public think about social safety net policies, as well as potentially, progressive taxation policies. I also think the proposed work is reported precisely and rigorously, and indeed, the study is well positioned for a RR format because we'd all be very interested in the results no matter what is found here. My comments, that I hope will be used to improve the manuscript, are below:

INTRO

I wonder if it makes sense to frame the gender hypotheses here more in terms of privilege v. masculinity. Or instead, it'd be better to talk about how both matter in the context of this research. Privilege matters because men have favored status in most societies and so their preference for wealth might be about a failure to acknowledge privilege or to deny advantage. However, male gender roles also would suggest a link between wealth and masculinity and so this pro-wealth bias may be less about privilege and more about masculinity. Right now, the privilege angle comes through but the gender role angle does not. I'm thinking of the precarious manhood work of vandello, the benevolent sexism work of Glick, Vescio, and Alice Eagly's considerable scholarship on this topic.

I'm with you on the hypothesis for the gender x income interaction but I get lost on the education piece in part because education is a multifaceted construct. With respect to wealth, some people learn how to accumulate wealth, so there is a learning component. There is also a status component as in, I'm yalie! There is also a culture component, being socialized around higher status people. As it stands I'm not sure which is operating in the 3-way interaction, and the argument here currently isn't precise enough about what education is and how it might relate to pro-wealth bias as a function of gender and income. All of this probably needs to be more precise so that the prediction becomes clearer. In fact, thinking on the different dimensions of education might lend itself to competing predictions!

ANALYSIS PLAN

I'm a little worried about statistical power for the three way interaction. I know the sims suggest that there is power, but I believe statistical power for interactions rests almost entirely on the shape of the interactions (data colada piece on now way interactions)--so if the pilot data is an overestimate of the true effect in the larger AIID data, you'll be under powered to test this three way interaction hypothesis. I'm not sure there is much you can do about this challenge in these data, but I do believe that the current power analyses are an overestimate, and you might consider simulating a range of power analyses to better get a sense of the uncertainty there. So if the shape is less extreme for the three-way interaction, what power do you have?

I am also wondering if the authors would consider analyzing some of the self-report data in their analysis plan. We have pro-wealth bias as DV here, but it would clarify what we find here if some of the self-report measures were also examined in a parallel regression analysis. For instance, I think there is information to be gained about understanding how much being rich/poor is consistent with the self or with felt cultural pressure. The parallel gender x income interaction with these measures might provide clues about what aspects of gender/education relate to pro-wealth bias. Without analyzing these data we may not learn as much as we could from these data. Of course, in my own exploring of these pilot data I think there are some limits to what people can analyze on the self-report end, but it's worth considering surely.

Also things don't "approach significance" ;)

Really interesting! Best of luck!

Editorial Office Comments to Authors:

It has come to our attention that the email address jcloutier@udel.edu is bouncing back into our system, either because the address is incorrect, or because the recipient has marked similar emails as spam. Before resubmitting your paper for consideration, we would be grateful if you could correct this error.

Author's Response to Decision Letter for (RSOS-191232.R0)

See Appendix A.

RSOS-191232.R1 (Revision)

Review form: Reviewer 1 (Benedict Jones)

Do you have any ethical concerns with this paper?

No

Recommendation?

Accept in principle

Comments to the Author(s)

This is a strong revision. In particular, the plans to include some analyses more akin to positive controls has really strengthened the protocol considerably. Thanks for taking the time to address each of my points.

Review form: Reviewer 2 (Michael Kraus)

Do you have any ethical concerns with this paper?

No

Recommendation?

Accept in principle

Comments to the Author(s)

I (Michael Kraus) feel really good about how the authors have thought through my and R1s prior comments during the previous round of review. I think we are better positioned to learn as much as we can from these data and that more of the appropriate context is included in the front end of the paper.

In terms of statistical power I guess my last thing is that I believe it isn't outside the realm of possible for the interaction effect size to be 30% lower even with this pilot sample (just by way of how data gets cut up in interactions) but this is more an intuition from working with data rather than a firm rule. I'm willing to concede given that other folks appear not to be worried.

I'm excited to see what the data reveal!

Decision letter (RSOS-191232.R1)

19-Feb-2020

Dear Dr Mattan,

On behalf of the Editor, I am pleased to inform you that your Manuscript RSOS-191232.R1 entitled "A registered report on how implicit pro-rich bias is shaped by the perceiver's gender and socioeconomic status" has been accepted in principle for publication in Royal Society Open Science. The reviewers' and editors' comments are included at the end of this email.

You may now progress to Stage 2 and complete the study as approved. Before commencing data collection we ask that you:

- 1) Update the journal office as to the anticipated completion date of your study.
- 2) Register your approved protocol on the Open Science Framework (<https://osf.io/>) or other recognised repository, either publicly or privately under embargo until submission of the Stage 2 manuscript. Please note that a time-stamped, independent registration of the protocol is mandatory under journal policy, and manuscripts that do not conform to this requirement cannot be considered at Stage 2. The protocol should be registered unchanged from its current approved state, with the time-stamp preceding implementation of the approved study design.

Following completion of your study, we invite you to resubmit your paper for peer review as a Stage 2 Registered Report. Please note that your manuscript can still be rejected for publication at Stage 2 if the Editors consider any of the following conditions to be met:

- The results were unable to test the authors' proposed hypotheses by failing to meet the approved outcome-neutral criteria.
- The authors altered the Introduction, rationale, or hypotheses, as approved in the Stage 1 submission.
- The authors failed to adhere closely to the registered experimental procedures. Please note that any deviations from the approved experimental procedures must be communicated to the editor immediately for approval, and prior to the completion of data collection. Failure to do so can result in revocation of in-principle acceptance and rejection at Stage 2 (see complete guidelines for further information).
- Any post-hoc (unregistered) analyses were either unjustified, insufficiently caveated, or overly dominant in shaping the authors' conclusions.
- The authors' conclusions were not justified given the data obtained.

We encourage you to read the complete guidelines for authors concerning Stage 2 submissions at <https://royalsocietypublishing.org/rsos/registered-reports#ReviewerGuideRegRep>. Please especially note the requirements for data sharing, reporting the URL of the independently registered protocol, and that withdrawing your manuscript will result in publication of a Withdrawn Registration.

Please note that Royal Society Open Science will introduce article processing charges for all new submissions received from 1 January 2018. Registered Reports submitted and accepted after this date will ONLY be subject to a charge if they subsequently progress to and are accepted as Stage 2 Registered Reports. If your manuscript is submitted and accepted for publication after 1

January 2018 (i.e. as a full Stage 2 Registered Report), you will be asked to pay the article processing charge, unless you request a waiver and this is approved by Royal Society Publishing. You can find out more about the charges at <https://royalsocietypublishing.org/rsos/charges>. Should you have any queries, please contact openscience@royalsociety.org.

Once again, thank you for submitting your manuscript to Royal Society Open Science and we look forward to receiving your Stage 2 submission. If you have any questions at all, please do not hesitate to get in touch. We look forward to hearing from you shortly with the anticipated submission date for your stage two manuscript.

Kind regards,
Lianne Parkhouse
Editorial Coordinator
Royal Society Open Science
openscience@royalsociety.org

on behalf of Professor Chris Chambers (Registered Reports Editor, Royal Society Open Science)
openscience@royalsociety.org

Associate Editor Comments to Author (Professor Chris Chambers):
Both reviewers are now satisfied and Stage IPA can be awarded.

Reviewers' comments to Author:

Reviewer: 1
Comments to the Author(s)

This is a strong revision. In particular, the plans to include some analyses more akin to positive controls has really strengthened the protocol considerably. Thanks for taking the time to address each of my points.

Reviewer: 2
Comments to the Author(s)

I (Michael Kraus) feel really good about how the authors have thought through my and R1s prior comments during the previous round of review. I think we are better positioned to learn as much as we can from these data and that more of the appropriate context is included in the front end of the paper.

In terms of statistical power I guess my last thing is that I believe it isn't outside the realm of possible for the interaction effect size to be 30% lower even with this pilot sample (just by way of how data gets cut up in interactions) but this is more an intuition from working with data rather than a firm rule. I'm willing to concede given that other folks appear not to be worried.

I'm excited to see what the data reveal!

Author's Response to Decision Letter for (RSOS-191232.R1)

See Appendix B.

RSOS-191232.R2 (Revision)

Review form: Reviewer 1 (Benedict Jones)

Is the manuscript scientifically sound in its present form?

Yes

Are the interpretations and conclusions justified by the results?

Yes

Is the language acceptable?

Yes

Do you have any ethical concerns with this paper?

No

Have you any concerns about statistical analyses in this paper?

No

Recommendation?

Accept with minor revision

Comments to the Author(s)

This is a strong registered report. Deviations from the stage one protocol are clear and justified. My only suggestion would be to fully report the exploratory analyses excluding participants under 18 years of age in the SI.

Review form: Reviewer 2 (Michael Kraus)

Is the manuscript scientifically sound in its present form?

Yes

Are the interpretations and conclusions justified by the results?

Yes

Is the language acceptable?

Yes

Do you have any ethical concerns with this paper?

No

Have you any concerns about statistical analyses in this paper?

No

Recommendation?

Accept with minor revision

Comments to the Author(s)

In this stage 2 reg report the authors have done an excellent job of completing their project using the AIID confirmatory dataset. Overall, I think the authors did a nice job completing the two stages of the RR process, I think the inferences are sound, the explanation of the results is clear,

and the paper will be an important data point for people thinking about pro-rich bias and its relation to gender and income and their associated stereotypes.

I suppose there are very small details that could be amended. In the methods there is a point where the paper talks about "this stage I report" and I thought that language could be amended even though it is technically a deviation from the prior report to more accurately reflect the completed project. Also, I thought the results could be organized in a slightly less cumbersome way where some of the exploratory analysis with multiple interaction terms could be relegated to a supplement. So things like education X income X gender analyses feel particularly exploratory and if you had a clear pattern I'm not sure (a) what it would be and (b) how I might explain it or not with a theoretical framework. But I leave this up to the authors.

Well done!
Michael Kraus

Decision letter (RSOS-191232.R2)

Dear Dr Mattan:

On behalf of the Editor, I am pleased to inform you that your Stage 2 Registered Report RSOS-191232.R2 entitled "A registered report on how implicit pro-rich bias is shaped by the perceiver's gender and socioeconomic status" has been deemed suitable for publication in Royal Society Open Science subject to minor revision in accordance with the referee suggestions. Please find the referees' comments at the end of this email.

The reviewers and Subject Editor have recommended publication, but also suggest some minor revisions to your manuscript. Therefore, I invite you to respond to the comments and revise your manuscript.

Please also ensure that all the below editorial sections are included where appropriate -- if any section is not applicable to your manuscript, please can we ask you to nevertheless include the heading, but explicitly state that the heading is inapplicable. An example of these sections is attached with this email.

- Ethics statement

- Data accessibility

It is a condition of publication that all supporting data are made available either as supplementary information or preferably in a suitable permanent repository. The data accessibility section should state where the article's supporting data can be accessed. This section should also include details, where possible of where to access other relevant research materials such as statistical tools, protocols, software etc can be accessed. If the data has been deposited in an external repository this section should list the database, accession number and link to the DOI for all data from the article that has been made publicly available. Data sets that have been

deposited in an external repository and have a DOI should also be appropriately cited in the manuscript and included in the reference list.

If you wish to submit your supporting data or code to Dryad (<http://datadryad.org/>), or modify your current submission to dryad, please use the following link:
[http://datadryad.org/submit?journalID=RSOS&manu=\(Document not available\)](http://datadryad.org/submit?journalID=RSOS&manu=(Document not available))

- **Competing interests**

- **Authors' contributions**

- **Acknowledgements**

- **Funding statement**

Because the schedule for publication is very tight, it is a condition of publication that you submit the revised version of your manuscript within 7 days (i.e. by the 02-Jul-2020). If you do not think you will be able to meet this date please let me know immediately.

- 1) A text file of the manuscript (tex, txt, rtf, docx or doc), references, tables (including captions) and figure captions. Do not upload a PDF as your "Main Document".

- 2) A separate electronic file of each figure (EPS or print-quality PDF preferred (either format should be produced directly from original creation package), or original software format)
- 3) Included a 100 word media summary of your paper when requested at submission. Please ensure you have entered correct contact details (email, institution and telephone) in your user account
- 4) Included the raw data to support the claims made in your paper. You can either include your data as electronic supplementary material or upload to a repository and include the relevant doi within your manuscript
- 5) All supplementary materials accompanying an accepted article will be treated as in their final form. Note that the Royal Society will neither edit nor typeset supplementary material and it will be hosted as provided. Please ensure that the supplementary material includes the paper details where possible (authors, article title, journal name).

Please note that Royal Society Open Science will introduce article processing charges for all new submissions received from 1 January 2018. Registered Reports submitted and accepted after this date will ONLY be subject to a charge if they subsequently progress to and are accepted as Stage 2 Registered Reports. If your manuscript is submitted and accepted for publication after 1 January 2018 (i.e. as a full Stage 2 Registered Report), you will be asked to pay the article processing charge, unless you request a waiver and this is approved by Royal Society Publishing. You can find out more about the charges at <https://royalsocietypublishing.org/rsos/charges>. Should you have any queries, please contact openscience@royalsociety.org.

on behalf of Professor Chris Chambers
(Registered Reports Editor, Royal Society Open Science)
openscience@royalsociety.org

Associate Editor Comments to Author (Professor Chris Chambers):

The Stage 2 manuscript was returned to the two reviewers who assessed it at Stage 1. Both reviewers are broadly satisfied, and both have recommendations for the reporting of the exploratory analyses (which may not be entirely compatible with each other). Under these circumstances, I am happy for the authors to decide how best to respond to these suggestions. Concerning the comment by Reviewer 2 on the use of the terminology "this stage-1 registered report" in the Method (p14 as part of the sentence "For this stage-1 registered report, we only have access to the smaller sample intended for pilot analyses"), I agree that this is best altered to avoid confusion as "At stage 1 of this registered report, we only had access to the smaller sample intended for pilot analyses". Provided the authors respond thoroughly to the points raised by the

reviewers, final Stage 2 acceptance will be forthcoming without requiring further in-depth review.

Comments to Author:

Reviewer: 1

Comments to the Author(s)

This is a strong registered report. Deviations from the stage one protocol are clear and justified. My only suggestion would be to fully report the exploratory analyses excluding participants under 18 years of age in the SI.

Reviewer: 2

Comments to the Author(s)

In this stage 2 reg report the authors have done an excellent job of completing their project using the AIID confirmatory dataset. Overall, I think the authors did a nice job completing the two stages of the RR process, I think the inferences are sound, the explanation of the results is clear, and the paper will be an important data point for people thinking about pro-rich bias and its relation to gender and income and their associated stereotypes.

I suppose there are very small details that could be amended. In the methods there is a point where the paper talks about "this stage I report" and I thought that language could be amended even though it is technically a deviation from the prior report to more accurately reflect the completed project. Also, I thought the results could be organized in a slightly less cumbersome way where some of the exploratory analysis with multiple interaction terms could be relegated to a supplement. So things like education X income X gender analyses feel particularly exploratory and if you had a clear pattern I'm not sure (a) what it would be and (b) how I might explain it or not with a theoretical framework. But I leave this up to the authors.

Well done!

Michael Kraus

Author's Response to Decision Letter for (RSOS-191232.R2)

See Appendix C.

Decision letter (RSOS-191232.R3)

Dear Dr Mattan:

It is a pleasure to accept your manuscript entitled "A registered report on how implicit pro-rich bias is shaped by the perceiver's gender and socioeconomic status" in its current form for publication in Royal Society Open Science.

You can expect to receive a proof of your article in the near future. Please contact the editorial office (openscience_proofs@royalsociety.org) and the production office

(openscience@royalsociety.org) to let us know if you are likely to be away from e-mail contact -- if you are going to be away, please nominate a co-author (if available) to manage the proofing process, and ensure they are copied into your email to the journal.

on behalf of Professor Chris Chambers (Subject Editor)
openscience@royalsociety.org

Appendix A

Editor Comments to Author (Professor Chris Chambers):

Two reviewers have now appraised the manuscript. Both are broadly positive but nevertheless identify a number of key issues that will need to be addressed in a revision. Reviewer 1 highlights areas where greater detail and rationale are required (e.g. concerning the exclusion criteria), and also notes the lack of outcome-neutral quality checks (e.g. positive controls). Reviewer 2 questions several aspects of the general rationale, the appropriateness of the power analysis, and the potential to report additional dimensions of the data. Overall, the issues raised appear readily addressable -- albeit substantial -- therefore a major revision is recommended.

We thank the reviewer for the opportunity to revise this registered report. We received the feedback with great interest and have endeavored to implement substantial changes in line with this feedback. We hope that these changes meet with approval and look forward to analyzing the hold-out dataset.

Comments to Author:

Reviewer: 1

This is quite far outside my own area, but the rationale for the study is very clearly specified and, as far as I can tell, well grounded in the literature. The hypotheses are also clearly specified and follow directly from previous work on the topic. Preliminary data is presented and related to them.

The analysis pipeline is generally very clear and appropriate, but several points would benefit from clarification, in my view.

P7. It wasn't initially clear to me that the rich-poor IAT was one of the 95 different topics that participants could be randomly assigned to.

We agree that this was not very clear in the original version of the manuscript. We have revised the draft to highlight this at the start of the Method section on page 13.

P8. Would be useful to list all the variables relevant to the confirmatory and exploratory analyses, clearly labeling which are relevant to which type of analysis.

We thank the reviewer for this excellent suggestion. The Measures section is now explicitly divided into separate sections that cover variables involved in confirmatory and exploratory analyses.

P9. I would mention here and archive the letter from Ebersole confirming you have not had access to the confirmatory portion of the data. (In general, a little more detail about the scheme through which you have accessed these data would be welcome, but probably not essential)

We think this is a great idea. The letter from Charlie Ebersole is now archived on the OSF at <https://osf.io/entbj/>. Additionally, on the OSF project page (https://osf.io/jcgyn/wiki/home/?view_only=fd598f1d2273492484d33a0d50c88050), we have

updated the project Wiki to describe the scheme through which we accessed the data. Information on the scheme for data access and the confirmation letter are provided on page 14 of the main text.

P9. It took me a few reads over to get what you meant by unique participants. Could clarify.

We agree that our use of this word was somewhat confusing in the last draft given that the meaning was only later clarified in the section on participant exclusions. To avoid any further confusion, we have removed the word “unique” from the sentence identified by the reviewer.

P9. Here (and in a couple of other places) you mention exclusion criteria being ‘in line with...’ recommendations made by others. It would be useful to be more precise. If you mean you will (or have) applied all those that they recommend, I’d list them specifically and state them more clearly.

This is a great idea. Were this suggestion more uniformly followed by the field, it would make replications much easier to conduct without sifting through referenced material that may not always make such criteria easy to find. To avoid confusing the reader with specifics of the IAT scoring algorithm prior to describing the IAT, we simply summarize which trial-level exclusions were implemented (see page 15). However, we do refer the reader to the section on IAT scoring (see pages 21-22), which now contains a thorough description of how D scores were computed, including any exclusions that are part of this process. Specifically, the AIID team excluded from analysis any trials with latencies exceeding 10,000 ms and replaced each error latency with its respective block’s average plus 600 ms. We confirmed with the AIID team that this was indeed the scoring algorithm that they used to compute the *D* scores provided in the AIID dataset.

P10. Same issue as previous point but for ‘following guidelines offered...’.

We have updated our description of participant-level exclusions to make it clear that Nosek and colleagues (2007) recommended criteria 1-7, but that we additionally adopted a final stricter criterion (see criterion 8 on page 16).

P9 and P10. It would be useful to give some justification for these exclusion criteria. Yes, they are being spelled out upfront (notwithstanding the points above), but a justification or explanation for each would be handy.

We agree that this would be helpful, particularly for readers who may be less familiar with the extensive literature that contributed to the procedures for analyzing IAT data. Accordingly, we have provided justification on page 15 for the trial-level exclusions recommended by Greenwald and colleagues (2003) that were also implemented in the present research. For the participant-level exclusion criteria listed on pages 15-16, Nosek and colleagues provide a collective justification for criteria 1-7 in footnote 4 of their 2007 paper. They write that these criteria are intended to exclude participants whose responses on the whole signal “careless performance”. We now include a note about this justification,

which served as the basis for Project Implicit's guidelines on participant-level exclusions. For criterion 8 (i.e., participants with $\geq 10\%$ RTs exceeding 10,000 ms in IAT critical blocks), we adopted it for similar reasons to criteria 1-7—namely, to exclude people who may have been insufficiently attentive during or confused by the IAT. We have opted to keep all exclusions in the present draft so as to minimize researcher degrees of freedom in this registered report. Nonetheless, if the reviewers and editor have strong reasons to amend this approach, we would be willing to abandon any of the above exclusions.

P11. You say it is difficult to know in advance how many participants will be in the confirmatory data set following exclusions. I'd argue it is impossible to know, so would recommend saying how many participants you will have prior to exclusion and approximately what proportion you'd expect to exclude based on your pilot analyses.

The reviewer is quite right that it is impossible to know in advance the final sample size of our confirmatory dataset. We have updated the section describing our projected sample size to reflect the reviewer's proposed approach (see page 17).

I really liked the power analysis by simulation and the assumptions made are fair (in my view).

We thank the reviewer for this positive assessment. We were excited to use this simulation-based approach; indeed, this is precisely the context where it is both easy and most useful to apply a simulation-based approach.

The level of detail in the methods and analysis pipeline is sufficient to replicate the work (but see points above where some clarifications are required).

Many thanks again to the reviewer for the feedback. We believe the methods and analysis pipeline is much clearer now.

Positive controls. My only major concern about this project is the lack of positive controls or manipulation checks. It might be that there are some form of attention checks in the original Project Implicit study that I am unaware of or some additional checks that could be added to the analysis pipeline (potentially as robustness checks?). It's hard for me to suggest any here, given I am not an expert with the IAT, but anything the authors can add to increase confidence in the procedures employed would be welcome (even if it was just an outline of robust findings replicated so far in Project Implicit, for example).

This is an excellent suggestion and we agree the project would benefit greatly from some positive controls. Before reporting on the results of some additional analyses below, we will first speak to previous findings that lend greater confidence in the present findings. The present finding of pro-rich bias in the IAT was anticipated based on a growing number of studies using implicit measures to assess attitudes about people with varying levels of social status. This includes work using wealth-based IATs (Cunningham, Nezlek, & Banaji, 2004; Horwitz & Dovidio, 2017; Rudman, Feinberg, & Fairchild, 2002), and our own recent work using status-based evaluative priming (Mattan et al., 2019), all of which show that high-status or rich people consistently elicit positive associations relative to their middle- and

low-status counterparts. Moreover, the rich-poor IAT tends to elicit substantially larger effect sizes relative to comparable IATs based on race, body weight, or religion (Rudman et al., 2002). Although many of these previous studies have explored effects of self-reported education, income, and subjective status on implicit status bias, results have so far been inconsistent with respect to whether these individual differences predict implicit bias. This may be in part due to insufficiently large sample sizes and relevant status characteristics of the perceiver such as minority status (see Experiment 2, Mattan et al., 2019) or gender. In the present experiment, we aim to address this shortcoming in a large online sample with a focus on perceiver gender, which—in some contexts—can be a group identity eliciting strong status associations (Berger, Cohen, & Zelditch, 1972; Eagly & Wood, 1982; Fiske, 1993).

Regarding positive controls, we did not have control over the experimental design for the AIID dataset. However, on further inspection of the dataset, we found many explicit measures of personal and perceived cultural attitudes toward the rich and the poor. These measures can help provide greater context on the sample's self-reported explicit attitudes and also allow for comparisons between self-report and implicit measures of attitudes. In this revision, we now report the analyses of these measures (see pages 35-38). In the final confirmatory dataset, we plan on conducting and reporting the same analyses to ensure that both samples show similar relationships.

Reviewer: 2

Comments to the Author(s)

I (Michael Kraus) think this is a really nice use of the AIID dataset. I hope my comments here are helpful as you prepare for the full analysis. Examining gender/income/education associations with pro wealth bias is an important consideration that will link up with how policy makers and the public think about social safety net policies, as well as potentially, progressive taxation policies. I also think the proposed work is reported precisely and rigorously, and indeed, the study is well positioned for a RR format because we'd all be very interested in the results no matter what is found here. My comments, that I hope will be used to improve the manuscript, are below:

Thank you, Prof. Kraus, for the positive feedback. We are also excited to learn about and report on what we find. Thanks to the insightful feedback from all reviewers, we believe the overall quality of this piece is greatly improved.

INTRO

I wonder if it makes sense to frame the gender hypotheses here more in terms of privilege v. masculinity. Or instead, it'd be better to talk about how both matter in the context of this research. Privilege matters because men have favored status in most societies and so their preference for wealth might be about a failure to acknowledge privilege or to deny advantage. However, male gender roles also would suggest a link between wealth and masculinity and so this pro-wealth bias may be less about privilege and more about masculinity. Right now, the privilege angle comes through but the gender role angle does not. I'm thinking of the precarious

manhood work of Vandello, the benevolent sexism work of Glick, Vescio, and Alice Eagly's considerable scholarship on this topic.

We agree that the impact of gender roles may also contribute to the effects we've observed so far in the pilot data. We have updated the introduction to feature some of the research underlying this additional motivation for our predictions. Indeed, the introduction now includes one section dedicated to the literature on gender roles, with a focus on external expectations toward both men and women. Another section focuses more on gender identity, with a focus on masculinity's unique association with status (e.g., precarious manhood hypothesis). Interestingly, both gender roles and gender identity mechanisms would lead to similar effects in the present analyses. Although participants who completed the rich-poor IAT as part of the AIID study did not complete measures of gender attitudes/beliefs that would allow us to parse the unique contributions from these mechanisms, we think it would be an important future direction to pursue.

I'm with you on the hypothesis for the gender x income interaction but I get lost on the education piece in part because education is a multifaceted construct. With respect to wealth, some people learn how to accumulate wealth, so there is a learning component. There is also a status component as in, I'm yalie! There is also a culture component, being socialized around higher status people. As it stands I'm not sure which is operating in the 3-way interaction, and the argument here currently isn't precise enough about what education is and how it might relate to pro-wealth bias as a function of gender and income. All of this probably needs to be more precise so that the prediction becomes clearer. In fact, thinking on the different dimensions of education might lend itself to competing predictions!

We thank the reviewer for this incisive comment. We agree it would be exciting to explore the contributions of different facets of education to the three-way interaction. Unfortunately, the AIID dataset includes only one variable that captures educational achievement. On the one hand, this constitutes a limitation of the present work that we intend to acknowledge in the discussion. On the other hand, the suggestion of considering these different dimensions of education when it comes to competing hypotheses is certainly something we can and should do even with the present dataset. Accordingly, we have articulated some competing predictions based on the different facets you have identified above. Specifically, we focus on the direct status-conveying component of education, particularly as it relates to the precarious manhood hypothesis (see pages 11-12). However, we also consider the cultural component of education by considering how differences in cultural fit may shape evaluations of the rich versus poor (see pages 10-11). These different components of education and their related theoretical accounts lead to partially distinct predictions, which we map out on page 13.

ANALYSIS PLAN

I'm a little worried about statistical power for the three way interaction. I know the sims suggest that there is power, but I believe statistical power for interactions rests almost entirely on the shape of the interactions (data colada piece on now way interactions)--so if the pilot data is an overestimate of the true effect in the larger AIID data, you'll be under powered to test this three way interaction hypothesis. I'm not sure there is much you can do about this challenge in these

data, but I do believe that the current power analyses are an overestimate, and you might consider simulating a range of power analyses to better get a sense of the uncertainty there. So if the shape is less extreme for the three-way interaction, what power do you have?

This is a valid concern. Although our pilot sample is perhaps larger than most pilot samples ($n = 175$), there is always a danger that pilot studies overestimate effect sizes, particularly when findings are somewhat novel. That being said, we like the suggestion of combining our simulation-based power analysis with a sensitivity approach, identifying the smallest effect size for the three-way interaction that our projected final sample could detect at 80% power. Assuming all other effect size parameters are equal and a desired power of 80%, these additional simulations indicated that a conservative sample size of 1,000 participants would be sufficient to detect a significant three-way interaction that is as much as 30% smaller than the effect size observed in our pilot sample. We now report the findings from this additional analysis on page 34.

I am also wondering if the authors would consider analyzing some of the self-report data in their analysis plan. We have pro-wealth bias as DV here, but it would clarify what we find here if some of the self-report measures were also examined in a parallel regression analysis. For instance, I think there is information to be gained about understanding how much being rich/poor is consistent with the self or with felt cultural pressure. The parallel gender x income interaction with these measures might provide clues about what aspects of gender/education relate to pro-wealth bias. Without analyzing these data we may not learn as much as we could from these data. Of course, in my own exploring of these pilot data I think there are some limits to what people can analyze on the self-report end, but it's worth considering surely.

We think this is a great idea, and it also helps address concerns raised by Reviewer 1. In addition to examining the relationship between IAT scores, perceiver status/gender predictors, and relevant self-report items, we also conducted additional parallel regressions looking at a selection of these self-report items. For these parallel regressions, we chose to focus on indices comprising multiple (rather than single) self-report items that assess differences related to attitudes toward the rich and the poor (see pages 23-27). Separately for the self, generic “others”, and society at large we analyzed (1) differences in explicit evaluations (poor minus rich), and (2) differences in felt pressure to adjust one’s evaluations (poor minus rich). This resulted in a total of six parallel models. We also included two additional models examining: (1) the difference in the degree to which accepting the poor (vs. rich) is important to the participant’s self-concept, and (2) the difference in ambivalence felt toward the rich (vs. poor). Ambivalence was computed separately for the rich and the poor as a weighted index of the minimum intensity of positive feelings and negative feelings divided by the difference in magnitude between positive and negative feelings.

Only two models showed any significant effects (see pages 39-42 for all parallel regressions). In the model examining the participant’s own attitudes toward the poor versus rich, we observed a significant Income \times Education interaction, $b = -0.5926$, $SE = 0.2159$, $CI_{95\%} = [-1.019, -0.166]$, $t(155) = -2.745$, $p = .007$. At low and average education levels, increasing income was associated with reduced explicit pro-rich bias, $b = 1.185$, $SE =$

0.399, $CI_{95\%} = [0.397, 1.972]$, $t(155) = 2.972$, $p = .003$. At high education levels, this pattern appeared to reverse, but this was non-significant, $b = -0.593$, $SE = 0.367$, $CI_{95\%} = [-1.318, 0.132]$, $t(155) = -1.616$, $p = .108$. At low income levels, increasing education was associated with reduced pro-rich bias, $b = 1.172$, $SE = 0.406$, $CI_{95\%} = [0.371, 1.973]$, $t(155) = 2.891$, $p = .004$. At high income levels, this pattern appeared to reverse, but this was non-significant, $b = -0.606$, $SE = 0.359$, $CI_{95\%} = [-1.316, 0.105]$, $t(155) = -1.684$, $p = .094$. In summary, the greatest pro-rich explicit bias was observed in individuals with the lowest income and education, which is the opposite of what we found in our analyses of IAT data. In addition to this Income \times Education interaction, we also observed a non-significant Gender \times Education interaction, $b = 0.791$, $SE = 0.410$, $CI_{95\%} = [-0.018, 1.601]$, $t(155) = 1.931$, $p = .055$. This non-significant interaction was characterized by an apparent decrease in pro-rich explicit bias with increasing education levels, but only for men, $b = 0.679$, $SE = 0.335$, $CI_{95\%} = [0.017, 1.341]$, $t(155) = 2.025$, $p = .045$, and not women, $b = -0.112$, $SE = 0.236$, $CI_{95\%} = [-0.578, 0.353]$, $t(155) = -0.476$, $p = .635$. All other effects in this analysis of the participant's self-reported evaluations of the poor versus rich were non-significant, $p > .07$.

In the model examining the participant's ambivalence toward the poor versus rich, we observed a significant main effect of education, $b = -0.084$, $SE = 0.040$, $CI_{95\%} = [-0.163, -0.005]$, $t(56) = -2.118$, $p = .039$. Greater education was associated with a stronger mixture of positive and negative feelings toward the rich compared to the poor. This is intriguing because greater education was also associated with generally greater pro-rich implicit bias. In other words, greater ambivalence may not necessarily correspond to a reduction in pro-rich implicit bias. In addition to this main effect of education, we also observed a significant Gender \times Income interaction, $b = 0.176$, $SE = 0.088$, $CI_{95\%} = [0.000, 0.352]$, $t(56) = 2.008$, $p = .050$. This interaction was characterized by a non-significant decline in ambivalence toward the rich (vs. poor) with increasing income for men, $b = 0.146$, $SE = 0.074$, $CI_{95\%} = [-0.002, 0.295]$, $t(56) = 1.972$, $p = .054$. This same relationship was non-significant for women, $b = -0.030$, $SE = 0.047$, $CI_{95\%} = [-0.123, 0.064]$, $t(56) = -0.636$, $p = .527$. This overall pattern is consistent with increasing pro-rich implicit bias as a function of increasing income in men but not women (see main analyses). All other effects in this analysis of ambivalence toward the rich versus the poor were non-significant, $p > .13$.

Like the other models, the self-concept model showed no significant effects. However, there were two non-significant main effects that may be worth commenting on. Should these main effects become reliable in the larger and therefore better powered confirmatory sample, they may provide additional evidence of internal pressure to modulate status bias in high-status individuals. Both gender (men $>$ women) and increasing income appeared to predict increasing agreement with the notion that accepting the poor (vs. rich) is important to one's self-concept, $b = 0.540$, $SE = 0.286$, $CI_{95\%} = [-0.025, 1.105]$, $t(56) = 1.889$, $p = .061$, and $b = 0.267$, $SE = 0.144$, $CI_{95\%} = [-0.018, 0.551]$, $t(56) = 1.850$, $p = .066$, respectively. Effects from all other models were non-significant, $p > .08$.

In conclusion, the parallel regression analyses of individual differences revealed relatively few effects in the pilot data. Among the few effects we observed, it would seem that increasing status is generally associated with diminished explicit pro-rich bias, but only for composite measures tapping into participants' own explicit evaluations of the poor and

rich. Although our sample for pilot analyses of attitudinal ambivalence was relatively small, we observed here a potentially important difference with respect to income and education. Whereas increasing education was associated with greater ambivalence toward the rich, increasing income was associated with reduced ambivalence toward the rich that was apparent only for men. In contrast with the findings for measures of individual attitudes, we observed no effects of external pressure to modulate class-based attitudes (cf. personal vs. others' and cultural attitudes). At this stage, our best interpretation of these results is that individuals with higher status (viz., through some combination of income, education, and/or gender) tend to downregulate their explicit positive evaluations of the rich, perhaps out of greater ambivalence among the educated and/or a desire to minimize apparent privilege (Kay & Jost, 2003) among the wealthy. Although we do not have the means to test the precise mechanism for status-based downregulation of pro-rich explicit bias in this dataset, we intend to follow up on these analyses in the main confirmatory dataset, providing the reader with (1) a replication of these piloted findings, (2) an extended discussion of how they may shed further light on the implicit/explicit divergence in pro-rich biases that occurs for high-status participants, and (3) additional discussion of future directions that may help uncover the mechanism underlying status-based downregulation of pro-rich bias.

Also things don't "approach significance" ;)

Thanks for catching that! We've changed the sentence to read as follows: "In our pilot analysis, the main effect of gender was non-significant albeit in the hypothesized direction..."

Really interesting! Best of luck!

Thank you!

Editorial Office Comments to Authors:

It has come to our attention that the email address jcloutier@udel.edu is bouncing back into our system, either because the address is incorrect, or because the recipient has marked similar emails as spam. Before resubmitting your paper for consideration, we would be grateful if you could correct this error.

Many thanks for letting us know about this. We have updated the email address, so this should hopefully not be an issue going forward.

Appendix B

College of Arts & Sciences

DEPARTMENT OF PSYCHOLOGICAL
& BRAIN SCIENCES

108 Wolf Hall
Newark, DE 19716-2577
Phone: 302-831-2571
Fax: 302-831-3645

August 21, 2020

Re: Stage-2 Registered Report Submission

Professor Christopher D. Chambers, BSc PhD CPsychol FBPsS
Registered Reports Editor – *Royal Society Open Science*

Dear Prof. Chambers:

Manuscript – A registered report on how implicit pro-rich bias is shaped by the perceiver’s gender and socioeconomic status (Mattan & Cloutier)

Accompanying this letter, please find our revised manuscript for your consideration as a stage-2 registered report for *Royal Society Open Science*. We want to thank you and the reviewers for the insightful feedback during the first stage of peer review.

Stage-2 analyses are now complete, and the manuscript has been updated accordingly. The final study repository containing study data, digital materials, and R analysis scripts is located at <https://osf.io/jcgyn/>. This link is referenced on page 29. The link for the stage-1 approved protocol is located at <https://osf.io/d5s23/> and cited in the manuscript on pages 12 and 30. As requested in the Royal Society Open Science author guidelines, we can confirm that no data other than pilot data included at stage 1 was subjected to pre-registered analyses prior to in-principle acceptance. We cited a letter confirming this in our stage-1 and stage-2 manuscripts (see page 14). That letter is archived online at <https://osf.io/entbj/>.

In this cover letter, I would like to take the opportunity to remind you of our email exchange from May 7, 2020 regarding a minor deviation from projected sample size. As we were drafting our stage-2 revision, we discovered that our original estimate of participants prior to exclusions erroneously included the pilot sample in the estimate total. The correct estimate should have been 1,553 participants prior to exclusions and 994 participants after exclusions. Fortunately, we based our power analysis off of a conservative projected sample size of 1,000 participants, which is pretty close to the correct projected final sample size, so those analyses will remain unchanged. That being said, our final sample size after exclusions was 767 participants, which is smaller than what our power analysis assumed, but still a large sample in its own right. The reason for this shortfall was because the number of participants prior to exclusions was lower than the correct projection of 1,553. In other words, the pilot dataset and confirmatory holdout dataset were about 17.8% and 82.2% of the total AIID data, respectively rather than the 15%/85% split we anticipated based on information from the AIID study team. In line with your helpful recommendations, we have documented this deviation in the method section (see footnotes 1–2 on page 17). In addition to reporting sensitivity analyses for the original projected sample size as we have already done, we have also reported sensitivity analyses for the final sample size (see page 35) which is consistent with our stated intentions in the accepted stage-1 manuscript.

As you may remember from the initial stage-1 submission, my co-author and I are submitting this manuscript in response to a call for registered reports using data from the Attitudes, Identities, and Individual Differences (AIID) dataset:

<https://docs.google.com/document/d/1zKqFrMzGsQga7XgBOwWGlmLI4aTfGo4T-xe-Hw-jP5A/edit>. Accompanying this submission, we also provide a letter from the AIID study team outlining the data-sharing agreement that my co-author and I signed. This letter is available at <https://osf.io/65stq/>. Because other research teams are independently using the same data, we are not able to share our confirmatory data until granted permission by the AIID study team. This letter is referenced in the manuscript under the data accessibility section on page 29 (see also page 14). Once we are given permission to publicly post our confirmatory dataset, we will do so on the project's repository on the Open Science Framework at <https://osf.io/jcgyn/>.

Please note that this registered report is not being reviewed in any other publication outlet and has not been offered in principle acceptance at any other outlet. Both authors have substantially contributed to this work and have approved the final version of this stage-2 manuscript. No potential or actual conflicts of interest exist in relation to the research reported. Data were collected in accordance with University of Virginia IRB protocol #2003017300. Analyses of this pre-existing dataset were ruled to be exempt from full review by the IRB at the University of Delaware, where the data were originally analyzed.

Sincerely,

Bradley D. Mattan, Ph.D.

Telephone: +1 (312)-450-9194

Email: brad.mattan@gmail.com

Appendix C

Dear Professor Chambers and reviewers,

We were pleased to hear that our registered report was recommended for publication, pending minor revisions. Your feedback throughout the peer review process has been invaluable, and we were eager to implement the remaining changes that you have suggested. We hope this most recent version adequately addresses the remaining concerns. We look forward to hearing from you and, as always, we are happy to implement any further feedback you may have on this work. We respond to all comments below.

Editorial Comments:

1) Please also ensure that all the below editorial sections are included where appropriate -- if any section is not applicable to your manuscript, please can we ask you to nevertheless include the heading, but explicitly state that the heading is inapplicable.

We have reviewed our revision to ensure that all sections are included and meet the editorial requirements: ethics statement (bottom of page 14), data accessibility statement (top of page 29), competing interests (page 49), author contributions (page 49), acknowledgements (page 49), and funding sources (page 49).

Associate Editor Comments to Author (Professor Chris Chambers):

The Stage 2 manuscript was returned to the two reviewers who assessed it at Stage 1. Both reviewers are broadly satisfied, and both have recommendations for the reporting of the exploratory analyses (which may not be entirely compatible with each other). Under these circumstances, I am happy for the authors to decide how best to respond to these suggestions. Concerning the comment by Reviewer 2 on the use of the terminology "this stage-1 registered report" in the Method (p14 as part of the sentence "For this stage-1 registered report, we only have access to the smaller sample intended for pilot analyses"), I agree that this is best altered to avoid confusion as "At stage 1 of this registered report, we only had access to the smaller sample intended for pilot analyses"). Provided the authors respond thoroughly to the points raised by the reviewers, final Stage 2 acceptance will be forthcoming without requiring further in-depth review.

We thank Prof. Chambers for facilitating a thorough and efficient peer review process and for the present guidance. In line with your and Prof. Kraus's helpful recommendations, we have updated the relevant passage on page 14 for clarity. It now reads as follows:

"At stage 1 of this registered report (see <https://osf.io/d5s23/>), we only had access to the smaller sample intended for pilot analyses (see <https://osf.io/entbj/> for confirmation letter from C. R. Ebersole)."

We outline our responses to the reviewers' remaining concerns below.

Comments to Author (Reviewer 1):

This is a strong registered report. Deviations from the stage one protocol are clear and justified. My only suggestion would be to fully report the exploratory analyses excluding participants under 18 years of age in the SI.

We agree that it would be useful to provide fuller details on our supplemental analysis after excluding participants under 18 years old. We have therefore created a supplemental document to accompany our registered report that now includes full details on this analysis in addition to some of the analyses that Prof. Kraus recommended we move to the supplemental document.

Comments to Author (Reviewer 2):

In this stage 2 reg report the authors have done an excellent job of completing their project using the AIID confirmatory dataset. Overall, I think the authors did a nice job completing the two stages of the RR process, I think the inferences are sound, the explanation of the results is clear, and the paper will be an important data point for people thinking about pro-rich bias and its relation to gender and income and their associated stereotypes.

We really appreciate your positive assessment of our work. We are also hopeful that others will find this registered report useful in thinking about implicit biases based on wealth (and status more broadly) and how they may be tied to gender, among other social categories.

I suppose there are very small details that could be amended. In the methods there is a point where the paper talks about "this stage I report" and I thought that language could be amended even though it is technically a deviation from the prior report to more accurately reflect the completed project.

Many thanks for catching this! We attempted to update such stage-1 anachronisms when revising from stage 1 to stage 2, but we missed this particular sentence. As outlined in our response to Prof. Chambers, we have updated the language on page 14.

Also, I thought the results could be organized in a slightly less cumbersome way where some of the exploratory analysis with multiple interaction terms could be relegated to a supplement. So things like education X income X gender analyses feel particularly exploratory and if you had a clear pattern I'm not sure (a) what it would be and (b) how I might explain it or not with a theoretical framework. But I leave this up to the authors.

This is a great suggestion. Particularly after adding in these exploratory analyses of the confirmatory dataset, this section became rather cumbersome, indeed. To streamline the manuscript, we have moved all results, tables, and figures for the parallel regressions to a supplemental document, leaving just the summaries of what we found for each dataset in the main text (see pages 38-39).